# Diversity-Preserving $K-$Armed Bandits, Revisited

**Hédi Hadiji**                                                                              *hedi.hadiji@centralesupelec.fr*
*L2S – CNRS – CentraleSupélec – Université Paris-Saclay, Gif-sur-Yvette*

**Sébastien Gerchinovitz**                                          *sebastien.gerchinovitz@irt-saintexupery.com*
*Institut de recherche technologique Saint Exupéry, Toulouse*
*Institut de mathématiques de Toulouse, Université Paul Sabatier, Toulouse*

**Jean-Michel Loubes**                                            *jean-michel.loubes@math.univ-toulouse.fr*
*Institut de mathématiques de Toulouse, Université Paul Sabatier, Toulouse*

**Gilles Stoltz**                                                        *gilles.stoltz@universite-paris-saclay.fr*
*Université Paris-Saclay, CNRS, Laboratoire de mathématiques d'Orsay, Orsay, France*
*HEC Paris, Jouy-en-Josas, France*

**Reviewed on OpenReview:** *https://openreview.net/forum?id=Viz7KBqO4A*

## Abstract

We consider the bandit-based framework for diversity-preserving recommendations introduced by Celis et al. (2019), who approached it in the case of a polytope mainly by a reduction to the setting of linear bandits. We design a UCB algorithm using the specific structure of the setting and show that it enjoys a bounded distribution-dependent regret in the natural cases when the optimal mixed actions put some probability mass on all actions (i.e., when diversity is desirable). The regret lower bounds provided show that otherwise, at least when the model is mean-unbounded, a $\ln T$ regret is suffered. We also discuss an example beyond the special case of polytopes.

## 1 Introduction

Ensuring fairness in recommendation systems has been a major concern in the machine learning literature among the last years, due to the growing influence of recommender systems in all parts of our societies. If many definitions of fairness have been provided, here we focus on models that ensure minimum allocation guarantees. Hence fairness has to be understood in terms of diversity-preserving constraints, as stated in the recent EU Artificial Intelligence Act[1], which imposes that AI recommender system should preserve the fundamental right of equal access to opportunities. We refer to Silva et al. (2022), Li et al. (2021), or Huang et al. (2021), and references therein, for some recent reviews on this topic. We will consider a bandit-based framework to model such recommendation algorithms, as introduced by the seminal work of Celis et al. (2019). The fairness framework presented in this study addresses a variety of scenarios involving the distribution of resources. In the context of search engine advertisements, it mandates that every advertiser is allocated a specified share of advertisement impressions, thereby preventing any single entity from dominating the available ad space. Similarly, in task planning platforms, it ensures that every participant receives a proportionate number of tasks, which is crucial to avoid disparate treatments. Furthermore, within wireless communication systems, this model obligates the receiver to maintain a baseline level of service quality for every transmitter, guaranteeing that all senders receive equitable treatment. This approach not only promotes fairness but also enhances the overall system's effectiveness by fostering diversity and participation across different settings. We refer to Patil et al. (2021) or Molina et al. (2023) for different methods promoting fairness in this context.

---

[1]https://artificialintelligenceact.eu/

**The setting by Celis et al. (2019).** We consider stochastic $K$–armed bandits. All arms correspond to desirable actions or options, though some lead to higher payoffs. Effective (regret-minimizing) algorithms are bound to play the optimal arm(s) an overwhelming fraction of time. Celis et al. (2019) refer to this effect as polarization (see details in Example 1) and introduce the learning protocol of Section 1.1 to avoid it. In a nutshell, this protocol consists of picking arms $A_t$ in a two-stage randomization, by choosing first a probability distribution $\underline{p}_t$ over the arms within a set $\mathcal{P}$ of admissible distributions, and then by drawing the arm $A_t$ to be actually played according to this distribution $\underline{p}_t$. The simplest example of admissible distributions is composed (see Example 1) of distributions all putting at least some minimal probability mass $\ell > 0$ on each arm. All actions or options corresponding to the arms then get some fair chance to be selected, even if their expected payoffs are far from the expected payoff of the optimal arms. We therefore suggest the alternative terminology of preserving diversity. The two-stage randomization considered makes sense in scenarios where there is a strong internal commitment to respect diversity but randomization is needed to respect privacy, or where some central authority eventually picks the arms based on the distributions output. Additional justifications of the setting are provided by Celis et al. (2019).

**Overview of our results.** Our aim in this article is to deepen the theoretical results obtained by Celis et al. (2019), see more details in Section 1.2. In a nutshell, Celis et al. (2019) approached the problem described above for polytopes $\mathcal{P}$ and by reducing it to the case of linear bandits: by ignoring that actions $A_t$ are eventually played and by only considering the distributions $\underline{p}_t$ chosen. Doing so, they obtain suboptimal rates for regret bounds. In particular, we show that bounded regret may be achieved in the favorable cases where diversity is actually desired: for sub-Gaussian models and when the optimal admissible distributions put some positive probability mass on all arms (which happens, in particular, when the set $\mathcal{P}$ is bounded away from the boundary of the simplex of probability distributions). Actually, at least in mean-unbounded sub-Gaussian models, we even characterize bounded regret as a feature of optimal distributions putting some probability mass on all arms. We also consider a case where $\mathcal{P}$ is not a polytope but a set with a continuous and curved set of extremal points. In that case, the reduction to linear bandits would provide $\sqrt{T}$ regret bounds while we show that using the information given by $A_t$, the regret may grow only at a squared-logarithmic rate, again showing a stark improvement.

A case of special interest (see Example 2) is a version of bandits with knapsacks where constraints are to be satisfied in expectation at all rounds, and thus, by martingale convergence, also almost surely in the limit. The associated theory is, of course, of a fundamentally simpler nature than the hard constraints typically considered in the literature of bandits with knapsacks (Badanidiyuru et al., 2013, Badanidiyuru et al., 2018).

## 1.1 The diversity-preserving setting introduced by Celis et al. (2019)

**Definition 1.** *A model $\mathcal{D}$ is a collection of distributions $\nu$ with finite first moments denoted by $\mu = \mathrm{E}(\nu)$.*

A $K$–armed bandit problem $\underline{\nu} = (\nu_1, \ldots, \nu_K)$, unknown to the learner, is fixed. At each round, the learner picks an arm $A_t \in [K]$, where $K = \{1, \ldots, K\}$, and obtains a payoff $Y_t$ drawn at random according to $\nu_{A_t}$ conditionally to the choice of $A_t$. This is the only observation made. So far, the setting described is exactly the one of vanilla $K$–armed bandits; the distinguishing feature of the bandit model by Celis et al. (2019) is that the choice of $A_t$ is actually made in two steps, as follows. First, a distribution $\underline{p}_t$ over $[K]$ is picked, in some known closed set $\mathcal{P}$ of the set $\mathcal{M}_{1,+}\big([K]\big)$ of all probability distributions over $[K]$. Distributions in the set $\mathcal{P}$ satisfy some diversity-preserving constraints (specific examples are given below). Then, the arm $A_t$ is drawn at random according to $\underline{p}_t$. Following game-theoretic terminology, we will call $a \in [K]$ pure actions or arms, and $\underline{p} \in \mathcal{P}$ mixed actions or distributions.

We measure performance in terms of expected payoffs. More precisely, denoting by $\mu_j = \mathrm{E}(\nu_j)$ the expectation of $\nu_j$ and by $\underline{\mu} = (\mu_1, \ldots, \mu_K)$ their vector, we first note that at round $t \geqslant 1$, by repeated applications of the tower rule,

$$\mathbb{E}\big[Y_t \mid A_t, \underline{p}_t\big] = \mu_{A_t}, \quad \text{thus} \quad \mathbb{E}\big[Y_t \mid \underline{p}_t\big] = \sum_{k \in [K]} p_{t,k}\, \mu_k \stackrel{\text{def}}{=} \big\langle \underline{p}_t, \underline{\mu} \big\rangle, \quad \text{thus} \quad \mathbb{E}\big[Y_t\big] = \mathbb{E}\big[\big\langle \underline{p}_t, \underline{\mu} \big\rangle\big]. \quad (1)$$

---

**Box A: Protocol of diversity-preserving stochastic bandits (Celis et al., 2019)**

**Known parameters**
- Arms $1, \ldots, K$ and model $\mathcal{D}$ of distributions for the arms
- Closed set $\mathcal{P} \subseteq \mathcal{M}_{1,+}\big([K]\big)$ of diverse enough probability distributions over the arms

**Unknown parameters**
- Probability distributions $\underline{\nu} = (\nu_1, \ldots, \nu_K)$ in $\mathcal{D}$, with expectations $\underline{\mu} = (\mu_1, \ldots, \mu_K)$

**For** $t = 1, 2, \ldots,$
1. Pick a distribution $\underline{p}_t = (p_{t,1}, \ldots, p_{t,K}) \in \mathcal{P}$ over the arms
2. Draw at random an arm $A_t \sim \underline{p}_t$
3. Get and observe a payoff $Y_t \sim \nu_{A_t}$ drawn at random according to $\nu_{A_t}$ given $A_t$

**Aim**
- Minimize the diversity-preserving expected regret $\qquad R_T = T \max_{\underline{p} \in \mathcal{P}} \langle \underline{p},\, \underline{\mu} \rangle - \mathbb{E}\left[ \sum_{t=1}^{T} \langle \underline{p}_t,\, \underline{\mu} \rangle \right]$

---

We consider expected regret: its distinguishing feature compared to vanilla $K$–armed bandits is that the performance of the learner is compared to distributions $\underline{p} \in \mathcal{P}$ only, not all distributions in $\mathcal{M}_{1,+}\big([K]\big)$. More precisely, we define the $\mathcal{P}$–diversity-preserving expected regret as

$$R_T = T \max_{\underline{p} \in \mathcal{P}} \langle \underline{p},\, \underline{\mu} \rangle - \mathbb{E}\left[ \sum_{t=1}^{T} \langle \underline{p}_t,\, \underline{\mu} \rangle \right].$$

The diversity-preserving expected regret is smaller than the vanilla expected regret for $K$–armed bandits, which corresponds to comparing to the maximum over $\underline{p} \in \mathcal{M}_{1,+}\big([K]\big)$. In some cases (see Theorem 1 and Corollary 1), the diversity-preserving expected regret may even be bounded.

The learning protocol and the regret goal are summarized in Box A.

**Example 1** (Avoiding polarization)**.** The main example by Celis et al. (2019) consists of imposing that each arm should be played at each round with some minimal probability $\ell > 0$, which corresponds to the diversity-preserving set $\mathcal{P} = \big\{ \underline{p} : \ \forall a \in [K], \ \ p_a \geqslant \ell \big\}$. This constraint makes sense in online advertisement: all offers need to be displayed a significant fraction of the time and get a significant chance to be selected. As argued by Celis et al. (2019), this wish of diversity would not take place with classical bandit algorithms, which would display almost only the most profitable ad (a phenomenon called polarization). More generally, for each arm $a$, depending on the contract passed, there could be a known range $[\ell_a, u_a]$ of individual probabilities of display, so that

$$\mathcal{P} = \left\{ \underline{p} : \ \forall a \in [K], \ \ p_a \in [\ell_a,\, u_a] \right\}.$$

**Example 2** (One-shot version of bandits with knapsacks in the mechanism design)**.** This second example is our own. Suppose that every pure action $a$ is associated with $N$ costs $c_a^{(1)}, \ldots, c_a^{(N)}$ in $\mathbb{R}$, accounting for limited resources or environmental costs like the amount of carbon emissions generated from taking the action; negative costs (e.g., negative carbon emissions) are allowed. When a player picks a pure action $A_t$ according to the mixed action $\underline{p} = (p_1, \ldots, p_K)$, the $N$ expected costs associated with her choice are

$$\sum_{a \in [K]} p_a c_a^{(1)}, \ldots, \sum_{a \in [K]} p_a c_a^{(N)}.$$

In this case, a reasonable objective for the player is to maximize her payoff under the constraints that, for all $n \in [N]$, the $n$–th expected cost of her actions be kept under a certain pre-specified level $u_n \in \mathbb{R}$. This

amounts to playing with the probability set

$$\mathcal{P} = \left\{ \underline{p} : \quad \forall n \in [N], \quad \sum_{a=1}^{K} p_a c_a^{(n)} \leqslant u_n \right\}.$$

Note that the name "diversity-preserving" was inspired by the example of the previous paragraph and is perhaps less relevant in the present example. The present example is a one-shot version of bandits with knapsacks (Badanidiyuru et al., 2013, Badanidiyuru et al., 2018), where the budget constraints must be satisfied at all rounds but in the mechanism design, and not when actual costs are summed over all rounds (though the latter will be approximatively achieved by martingale convergence).

In both examples, the diversity-preserving sets $\mathcal{P}$ considered are polytopes in the following sense.

**Definition 2.** *A polytope $\mathcal{P}$ is a convex set generated by finitely many extremal points whose set is denoted by* $\mathrm{Ext}(\mathcal{P})$.

When $\mathcal{P}$ is a polytope, we may introduce suboptimality gaps of distributions in $\mathrm{Ext}(\mathcal{P})$ and decompose the diversity-preserving regret as stated in (2). More precisely, we denote by $\mathrm{Opt}(\underline{\nu}, \mathcal{P})$ the subset of optimal distributions, which achieve an expected payoff denoted by $M(\underline{\mu}, \mathcal{P})$, i.e.,

$$M(\underline{\mu}, \mathcal{P}) = \max_{\underline{p} \in \mathcal{P}} \langle \underline{p}, \underline{\mu} \rangle \qquad \text{and} \qquad \mathrm{Opt}(\underline{\nu}, \mathcal{P}) = \operatorname*{argmax}_{\underline{p} \in \mathcal{P}} \langle \underline{p}, \underline{\mu} \rangle,$$

and define in turn the suboptimality gap $\Delta_{\underline{p}}$ of a given distribution $\underline{p} \in \mathrm{Ext}(\mathcal{P})$, and the global suboptimality gap $\Delta$ of the set $\mathrm{Ext}(\mathcal{P})$, as

$$\Delta_{\underline{p}} = M(\underline{\mu}, \mathcal{P}) - \langle \underline{p}, \underline{\mu} \rangle \qquad \text{and} \qquad \Delta = \min\big\{ \Delta_{\underline{p}} : \ \underline{p} \in \mathrm{Ext}(\mathcal{P}), \ \Delta_{\underline{p}} > 0 \big\}.$$

(We assume that at least one $\underline{p} \in \mathrm{Ext}(\mathcal{P})$ is such that $\Delta_{\underline{p}} > 0$, otherwise, all strategies have a null regret.) Denoting by $N_{\underline{p}}(T)$ the number of times a distribution $\underline{p} \in \mathrm{Ext}(\mathcal{P})$ is played, the diversity-preserving regret of a (possibly randomized) strategy only picking distributions $\underline{p}_t \in \mathrm{Ext}(\mathcal{P})$ can then be rewritten as

$$R_T = T \max_{\underline{p} \in \mathcal{P}} \langle \underline{p}, \underline{\mu} \rangle - \mathbb{E}\left[ \sum_{t=1}^{T} \langle \underline{p}_t, \underline{\mu} \rangle \right] = \sum_{\underline{p} \in \mathrm{Ext}(\mathcal{P}) \backslash \mathrm{Opt}(\underline{\nu}, \mathcal{P})} \Delta_{\underline{p}} \, \mathbb{E}\big[ N_{\underline{p}}(T) \big], \quad \text{where} \quad N_{\underline{p}}(T) = \sum_{t=1}^{T} \mathbb{1}_{\{\underline{p}_t = \underline{p}\}}. \quad (2)$$

Actually, Celis et al. (2019) consider the possibility of contexts in their description of the setting, but then introduce and analyze policies that "function independently for each context". The setting and aim described above and summarized in Box A correspond exactly to the framework of Theorems 1 and 2 of Celis et al. (2019).

## 1.2 Overview of our results and comparison to Celis et al. (2019)

Celis et al. (2019) provide some extensive numerical studies but we only discuss below their theoretical contributions; the latter only dealt with diversity-preserving sets $\mathcal{P}$ given by polytopes.

**Regret upper bounds by Celis et al. (2019): for polytopes.** They first suggest using linear-bandit algorithms in the diversity-preserving setting, since (1) indicates that the expected reward is a linear function of the played distribution $\underline{p}_t$. Of course, with such a reduction, the learner discards a some useful information: the value $A_t$ of the arm actually played. This is why Celis et al. (2019) obtained suboptimal regret bounds. More precisely, they use the LinUCB strategies (linear upper confidence bound) strategies introduced by Li et al. (2010) and Chu et al. (2011) and further studied by Abbasi-Yadkori et al. (2011). With this strategy, they obtain regret bounds of order $K(\ln T)^2/\Delta$. A second suggestion by Celis et al. (2019) is a constrained version of the $\varepsilon$–greedy algorithm by Auer et al. (2002), which does use the knowledge of the arms $A_t$ actually played and obtains a regret bound of order $(\ln T)/\Delta^2$. However, this constrained $\varepsilon$–greedy algorithm requires the value of $\Delta$ to tune its exploration rate $\varepsilon_t$ over time; this limitation makes it impractical compared to knowledge-independent algorithms.

For both strategies of Celis et al. (2019), the case of a bounded regret is not covered, while it constitutes our main contribution.

**Regret upper bounds for polytopes in this article: Section 2 and Appendix A.** More precisely, we introduce in Section 2 a diversity-preserving UCB strategy, close to the original UCB strategy by Auer et al., 2002: it maintains indexes on arms and outputs the best distribution in $\mathcal{P}$ given these indexes. Additional minor modifications are needed as the strategy cannot ensure, unlike in the classic case, that all arms are first pulled once. For sub-Gaussian models and for diversity-preserving sets $\mathcal{P}$ given by polytopes, this diversity-preserving UCB strategy enjoys a bounded regret as soon as all optimal distributions for a problem put some positive probability mass on each arm; this condition somehow indicates that each arm is desirable and flags cases where diversity is welcome. Even when the condition is not met, a $\ln T$ regret is suffered.

Both regret bounds are stated in Theorem 1, whose proof may be found in Appendix A. The proof of the $\ln T$ bound relies mostly on typical optimistic techniques for $K$–armed bandits, with occasional twists, to provide controls on the numbers of times arms $a \in [K]$ are played through information on the numbers of times distributions $\underline{p} \in \mathrm{Ext}(\mathcal{P})$ were picked. The proof of the bounded regret follows a completely different logic: we first show that optimal distributions are typically played at least half of the time, which, entails, because $p^{\star}_{\min}(\underline{\nu}) > 0$, that each pure action $a \in [K]$ is played linearly many times. Therefore, all estimates are sharp, and little regret is suffered. The diversity-preserving setting has it here that all arms may be, and even should be, played linearly many times—unlike in the vanilla $K$–armed bandit settings, where suboptimal arms are only played about $\ln T$ times.

Regarding computational complexity, the algorithm we propose only requires solving one linear program over the constraint set at every round, which is often less expensive than the double maximization over an ellipsoid of LinUCB (see Li et al., 2010, Chu et al., 2011, and Abbasi-Yadkori et al., 2011).

**Regret lower bounds in this article: Sections 3 and Appendices B–C.** Celis et al. (2019) do not provide lower bounds. We provide optimality results in Section 3 (with proofs located in Appendices B and C). They are stated under the restriction that the bandit problem $\underline{\nu}$ considered has a unique optimal distribution $\underline{p}^{\star}(\underline{\nu})$. Theorem 2 states that in models with no upper bound on the expectations (and satisfying two other mild technical conditions), all reasonable strategies (i.e., achieving a small diversity-preserving regret, for all bandit problems) must suffer a $\ln T$ when $\underline{p}^{\star}(\underline{\nu})$ puts no probability mass on at least one arm. Together with the upper bounds results discussed so far, we therefore obtain in Corollary 1 a characterization of the bounded versus $\ln T$ regrets for these models with no upper bound on the expectations.

The case of models with bounded expectations is more delicate and we illustrate in Proposition 1, with Bernoulli bandit problems, why stating such a characterization is challenging.

The proof techniques rely to a large extent on classic proof schemes for bandit lower bounds (Lai & Robbins, 1985, Graves & Lai, 1997, Garivier et al., 2019). The main difference raised by our setting is described in Remark 1: eventually, arms $A_t$ are played, so that the Kullback-Leibler information gain should be quantified in terms of the $A_t$ and is larger than if it was quantified in terms of the distributions $\underline{p}_t$ used; yet, the small-regret constraints on the strategies are in terms of the distributions $\underline{p}_t$ used. This leads to some specific constrained minimum on $\liminf R_T / \ln T$, à la Graves & Lai (1997), which is to be studied.

**Regret upper bounds for $\mathcal{P}$ given, e.g., by a ball: Section 4 and Appendix D.** We study a case where the diversity-preserving set $\mathcal{P}$ is not given by a polytope but whose set of extremal points is continuous and curved. For simplicity, we consider Euclidean balls in the probability simplex. We obtain a regret bound of order $\ln^2 T$ when such a ball $\mathcal{P}$ lies in the relative interior of the simplex. The striking feature of this bound is that a linear-bandit algorithm discarding the arms $A_t$ actually played would pay a regret of order at least $\sqrt{T}$, showing again a large gap between knowledge-informed algorithms and generic linear-bandit bounds when all actions are desirable.

## 1.3 Literature review

The many existing notions of algorithmic fairness have lead to different fair bandit problems formulations see, e.g, the contributions by Wang et al. (2021), Liu et al. (2017), Barman et al. (2023), which are unrelated to the setting we study. We focus our literature review below on comparable work.

We mention in passing the first version of the present article: Hadiji et al. (2020).

**Diversity.** Chen et al. (2020) study a particular case of the diversity-preserving setting, which essentially corresponds to Example 2. While their framework is the same as ours, they only study distribution-free bounds, at best of order $\sqrt{KT}$. Their algorithm applies to rewards generated in an adversarial manner.

In the domain of stochastic bandits, Patil et al. (2021) and Claure et al. (2020) design bandit algorithms ensuring that the proportion of times each action is selected is lower bounded, i.e., with our notation, that $N_a(T)/T \geqslant u_a$ at every round. This is similar in spirit to the special case of our setting presented in Example 1. However, the hard constraint on the number of pulls poses algorithmic design issues different from ours, which they solve only in the special case of lower bounds on every arm pull. Our setting is more flexible yet enforces similar guarantees while bypassing these issues.

After a preprint version (Hadiji et al., 2020) of the present article was published online, Liu et al. (2022), building on Li et al. (2020), considered a combinatorial bandit problem with constraints on the distribution over actions selected by the player. While they formally allow the player to play outside the constraint set, they measure regret with respect to the best constrained distribution, and their algorithm never plays outside the constraint set, making their setting essentially identical to ours. They analyze a natural extension of diversity-preserving UCB to the combinatorial setting, providing logarithmic regret bounds similar to our Theorem 1, and discuss specific cases in which constant regret can be achieved, e.g., when *all* distributions put some given positive probability on all actions. Compared to their results, we provide a *characterization* of when constant regret happens—that goes way beyond the case of a positive lower bound on the components of the distributions. Anecdotally, our upper bound on the constant regret, when specified to Example 1, is tighter than theirs. Indeed, the main term in our bound is of order $K/\bigl(\Delta\, p^\star_{\min}(\nu)\bigr)$ up to log terms, whereas they get a bound of order $1/\bigl(\Delta\, (p^\star_{\min}(\nu)^2)\bigr)$, which is always larger since $p^\star_{\min}(\nu) \leqslant 1/K$.

Note that all these articles refer to "fairness", although we use the term "diversity-preserving" in this case.

**Bounded regret in structured bandits.** A recent line of work (Hao et al., 2020, Tirinzoni et al., 2020, Jun & Zhang, 2020), pioneered by Bubeck et al. (2013) and Lattimore & Munos (2014), studies the possibility of achieving bounded regret in structured bandits. There, the learner knows a priori a structural property of the relationship between the payoffs of different actions available. Certain types of structures give rise to the "automatic exploration" phenomenon: despite the bandit feedback, playing the optimal arm (and knowing the structure) yields enough information to differentiate it from the suboptimal arms without playing them; this opens up the possibility of obtaining bounded regret.

These results are deeply connected to the lower bound of Graves & Lai (1997), which often provides the tight constant in front of the logarithmic asymptotic rate of regret in structured bandits, matched in wide generality by Combes et al. (2017); see also the equivalent formulation for linear bandits with Gaussian noise in Lattimore & Szepesvári (2017). Constant regret can only occur in problems for which that lower bound is 0.

As in most bandit settings, these works all assume that the learner has a deterministic control over the arms $A_t$ eventually picked. The diversity-preserving framework contrasts this aspect, by preventing the learner from picking $A_t$ and introducing some additional random draw of $A_t$ based on the distribution $\underline{p}_t$ picked. The relationship between the policy and the information acquired is thus altered in a central way. On a technical level, standard bandit algorithms may decrease mechanically the radius of the confidence bounds, whereas the extra randomness between $\underline{p}_t$ and $A_t$ only allows for some probabilistic control in the diversity-preserving setting. In particular, the analysis techniques of Lattimore & Munos (2014) and Tirinzoni et al. (2020), based on confidence bounds, need to be sharply refined to be used in the diversity-preserving setting.

Also, to the best of our knowledge, the results of Section 4 provide the first example of a structured bandit problem with an action set (the diversity-preserving set $\mathcal{P}$) with a continuous set of extremal points and for which automatic exploration occurs and brings the rate of regret from $\sqrt{T}$ down to $\ln^2 T$.

Note that automatic exploration also happens organically in Degenne et al. (2018), where the authors assume directly that the learner may observe the payoff of an extra arm.

**Mediator feedback.** The diversity-preserving setting of this article has notable connections with bandits with mediator feedback, on the one hand, and with regret minimization with expert advice, on the other hand. In all three settings, arms are not directly pulled but a distribution (possibly non-stationary) is placed in the middle.

Inspired by policy optimization in reinforcement learning, Metelli et al. (2021) introduce the problem of bandits with mediator feedback, which a is more general version of the diversity-preserving problem we consider. They define bandits with mediator feedback as any bandit problem in which the player observes an intermediate (possibly random) feedback $o_t$ that determines the reward. (Precisely, the reward $Y_t$ is independent of the action $A_t$ conditionally on $o_t$.) The authors analyze, in particular, the case of policy optimization within a class of policies parameterized by some $\theta \in \Theta$; the goal of the learner is to select policies (parameters) that yield high returns. In this example, the player observes the entire sequence of rewards and actions throughout the episode, which is more informative than the sole cumulative gain of the policy. Our setting is more specific but our analysis is also more precise. The results by Metelli et al. (2021) can indeed be applied to our setting, by considering a Markov Decision Process with a single state and by identifying the policies directly with distributions over the action set. Based on this reduction, their analysis does not make full use of the simple information structure available, which results both in suboptimal lower and upper bounds. More precisely, the regret bound of their algorithm becomes infinite as soon as $\mathcal{P}$ is fully not contained in the relative interior of the simplex, while our regret bounds are at worst of order $\ln T$. (In their notation, $\underline{p}$ corresponds to $\theta$, and $v(\theta) = \infty$.) On the lower bound side, they only provide a worst-case result, by exhibiting hard instances on which algorithms incur a minimal regret, whereas we characterize the optimal rates on every instance.

Another article worth to be mentioned is by Eldowa et al. (2023), in the context of regret minimization with expert advice: indeed, the finite policy set $\Theta$ that they consider in $K$–armed bandits could correspond to the set $\text{Ext}(\mathcal{P})$, which makes the setting studied here a special case of theirs with i.i.d. stochastic losses. However, to the best of our reading, we found no trace of constant regret and only bounds scaling with $\sqrt{T}$. This is due to the adversarial approach followed; their angle is to improve the regret bounds in terms of their dependencies on other quantities than $T$ (e.g., number of policies, arms, etc.), which they do through the information-theoretic quantities introduced.

**Extensions to Markov decision processes.** Occurrences of constant regret have also recently appeared in the reinforcement learning literature, in episodic settings more or less related to ours. For linear Markov Decision Processes, Zhang et al. (2024) derive algorithms that achieve constant regret with high probability, but with potentially large expected regret, and Papini et al. (2021) extend the work of Hao et al. (2020), discussed above, providing a necessary and sufficient for constant regret. Vera et al. (2021) introduce an online version of knapsack problems that accounts for the computational tractability of the benchmark strategies, and design constant regret algorithms against these benchmarks. Wagenmaker & Foster (2023) tackle the general and difficult question of characterizing the non-asymptotic instance-dependent optimal regret in reinforcement learning (contrasting with the asymptotic nature of the lower bounds following approaches à la Graves & Lai, 1997). Therein, constant-regret instances play a special role, since the constants, which are typically second-order terms, form the dominant term in the regret.

Note the difference with the setting of Metelli et al. (2021) discussed earlier: in their formulation of policy optimization, the action set corresponds to the set of parameters, and not to the action set of the MDP as in the references right above.

## 2 A simple diversity-preserving UCB strategy

**Definition 3.** *A distribution $\nu$ over $\mathbb{R}$ with expectation $\mu = \mathbb{E}(\nu)$ is $\sigma^2$–sub-Gaussian if*

$$\forall \lambda \in \mathbb{R}, \qquad \int e^{\lambda(x-\mu)} \, d\nu(x) \leqslant e^{\lambda^2 \sigma^2/2} \,.$$

We assume that the model $\mathcal{D}$ is composed of $\sigma^2$–sub-Gaussian distributions, where the parameter $\sigma^2$ is known. A lemma by Hoeffding (1963) indicates that the model $\mathcal{D}_{[a,b]}$ of all probability measures supported

---

**Box B: Diversity-preserving UCB for polytopes**

**Inputs**: sub-Gaussian parameter $\sigma^2$ for distributions in $\mathcal{D}$; polytope $\mathcal{P}$

**Initialization**: pick some $u_0$ taken by some distribution in $\mathcal{D}$, and let $\underline{U}(0) = (u_0, \dots, u_0)$

**For** rounds $t = 1, 2, \dots,$

    1. Select (ties broken arbitrarily) a distribution      $\underline{p}_t \in \underset{\underline{p} \in \text{Ext}(\mathcal{P})}{\text{argmax}} \left\langle \underline{p}, \underline{U}(t-1) \right\rangle$

    2. Play the pure action $A_t \sim \underline{p}_t$

    3. Get and observe the reward $Y_t \sim \nu_{A_t}$

    4. Compute the empirical averages

$$\widehat{\mu}_a(t) = \begin{cases} u_0 & \text{if } N_a(t) = 0, \\ \dfrac{1}{N_a(t)} \displaystyle\sum_{s=1}^{t} Y_s \mathbb{1}_{\{A_s = a\}} & \text{if } N_a(t) \geqslant 1, \end{cases} \quad \text{where} \quad N_a(t) = \sum_{s=1}^{t} \mathbb{1}_{\{A_s = a\}}$$

    5. Compute the upper confidence bound vector $\underline{U}(t) = \big( U_1(t), \dots, U_K(t) \big)$ according to

$$\forall a \in [K], \qquad U_a(t) = \widehat{\mu}_a(t) + \sqrt{\frac{8\sigma^2 \ln t}{\max\{N_a(t), 1\}}}$$

---

on a compact interval $[a, b]$ satisfies this assumption, with $\sigma^2 = (b-a)^2/4$. Of course, the model $\mathcal{D}$ of all Gaussian distributions with variance smaller than some prescribed $\sigma^2$ is also suitable.

We also impose for now that $\mathcal{P}$ is a polytope. See Section 4 for a case where $\mathcal{P}$ is not a polytope.

We then introduce a UCB strategy in Box B, which maintains indexes $U_a(t)$ directly and separately on the arms $a \in [K]$, as the classic UCB strategy. The strategy neither uses indexes for distributions in $\text{Ext}(\mathcal{P})$, nor resorts to some global linear-bandit estimation as in Abbasi-Yadkori et al. (2011). The indexes are initialized at some value $u_0$ possibly taken by some distribution in $\mathcal{D}$, and are of the form, for $t \geqslant 1$ and $a \in [K]$,

$$U_a(t) = \widehat{\mu}_a(t) + \sqrt{\frac{8\sigma^2 \ln t}{\max\{N_a(t), 1\}}},$$

where $N_a(t)$ counts the number of times arm $a$ was pulled till round $t$ included, and where $\widehat{\mu}_a(t)$ is the empirical payoff achieved over those rounds when $a$ was played. Of course, it may be that $N_a(t) = 0$, as the learner has no direct control on which arm is played—the learner cannot ensure that arm $a$ be picked even once. This is why we use $\max\{N_a(t), 1\}$ in the confidence bonus, and also, set $\widehat{\mu}_a(t) = u_0$ in the case $N_a(t) = 0$.

We prove (in Appendix A) the regret bounds stated below in Theorem 1.

**Theorem 1.** *Let $\mathcal{P}$ be a polytope. Consider a sub-Gaussian model $\mathcal{D}$ with parameter $\sigma^2$, known and used by the diversity-preserving UCB strategy of Box B. The regret of the latter satisfies*

$$\forall \underline{\nu} \text{ in } \mathcal{D}, \quad \forall T \geqslant 1, \qquad R_T \leqslant C_{\underline{\nu}} \ln T + c_{\underline{\nu}},$$

*where $C_{\underline{\nu}}$ and $c_{\underline{\nu}}$ are quantities depending on $\underline{\nu}$ and whose general closed-form expressions may be read in (12). In addition, for problems $\underline{\nu}$ in $\mathcal{D}$ such that*

$$p_{\min}^{\star}(\underline{\nu}) \overset{\text{def}}{=} \min_{\underline{p} \in \text{Opt}(\underline{\nu}, \mathcal{P})} \min_{a \in [K]} p_a > 0, \qquad \textit{the regret is even bounded:} \qquad \limsup_{T \to \infty} R_T < +\infty \, ;$$

*similarly, a closed-form finite upper bound may be read in (19).*

Some comments are in order, first on the cases of interest (and those that are mere sanity checks), and second, on the proof techniques.

**Cases of interest in Theorem 1.** The $\ln T$ bound in Theorem 1 could also be achieved (with the same computational complexity) by a UCB strategy taking the distributions in $\text{Ext}(\mathcal{P})$ as arms. The main achievement in Theorem 1 is therefore the case of bounded regret. The condition $p^\star_{\min}(\underline{\nu}) > 0$ corresponds to cases where diversity is indeed desirable, and this is when bounded regret is achieved.

The diversity-preserving UCB strategy of Box B coincides with the classic UCB strategy by Auer et al. (2002) in the case where $\mathcal{P}$ is the entire simplex of probability distributions. However, the general $\ln T$ bound achieved in Theorem 1, with closed-form expression (12), does not coincide with the classic UCB bound of Auer et al. (2002) in this case. This is due to some general but loose analysis followed in the proof of Theorem 1. We recall, once again, that the $\ln T$ part of Theorem 1 only forms some sanity check, the main achievement being the bounded-regret case.

**Intuition of the proof and proof techniques.** The proof of Theorem 1 is provided in Appendix A; actually, two separate proofs with two different logics are provided: one for the general $\ln T$ rate in Appendix A.1; one for the case of bounded regret in Appendix A.2.

The proof of the $\ln T$ bound was obtained by an adaptation of the classic UCB proofs, for vanilla $K$–armed bandits (see Auer et al., 2002) or for linear bandits (see, e.g., Abbasi-Yadkori et al., 2011 and Lattimore & Szepesvári, 2017). In particular, the proof shows, in view of (2), that suboptimal distributions $\underline{p} \in \text{Ext}(\mathcal{P})$ are unlikely to be played more than $\ln T$ times. We highlight in Appendix A the (three specific) technical modifications with respect to these classic proofs: they all basically amount to controlling, with high probability, the $N_a(t)$ based on the numbers of times $N_{\underline{p}}(t)$ distributions $\underline{p}$ are played.

The proof for the case of bounded regret follows a completely different logic (the challenge to be overcome here was to get and write down this alternative logic). We first show that optimal distributions are typically played at least half of the time. This entails, because $p^\star_{\min}(\underline{\nu}) > 0$, that each pure action $a \in [K]$ is played linearly many times. Therefore, all estimates are sharp, and little regret is suffered. This proof thus provides the rationale for the bounded regret: there is less competition than in the classic case between exploration and exploitation—exploitation involves exploration "by design" in the case where $p^\star_{\min}(\underline{\nu}) > 0$. This rationale is different from the one at hand in structured bandits (see Section 1.3 for the references): therein, arms played also provide some information on arms not played thanks to the underlying structure.

## 3 Lower bounds / Optimality in the case of polytopes

In this section, we discuss the situations where the $\ln T$ rate stated in Theorem 1 is unavoidable: does this happen when optimal distributions put no probability mass on at least one arm? To answer the question, we first provide in Section 3.1 (more precisely, in Theorem 2) a $\ln T$ lower bound on the regret against all bandit problems with a unique optimal arm not putting any probability mass on some arm, provided that the model is mean-unbounded in the sense of Definition 4 below.

In Section 3.2, we the combine the lower bound of Theorem 2 and the upper bound of Theorem 1 to show that for sub-Gaussian mean-unbounded models, either the regret may be bounded or it must grow at an optimal $\ln T$ rate. The case of mean-bounded models is more delicate and Proposition 1 illustrates the intrinsic difficulties in characterizing the rates for these models.

Before we proceed, we state our key distinction between mean-bounded and mean-unbounded models.

**Definition 4.** *A model $\mathcal{D}$ is said to be mean-bounded if there exists $M \in \mathbb{R}$ such that the expectation $\mu = \text{E}(\nu)$ of any $\nu \in \mathcal{D}$ satisfies $\mu \leqslant M$. Otherwise, the model $\mathcal{D}$ is said mean-unbounded: for all $M \in \mathbb{R}$, there exists $\nu \in \mathcal{D}$ such that $\mu = \text{E}(\nu) > M$.*

**Example 3.** *The rewards in online advertisement (see Example 1) are naturally bounded. Admittedly, most real applications are associated with mean-bounded models; mean-unbounded models (that are also sub-Gaussian) are less common. We could think of the model with distributions of the form: some mean plus some fixed, possibly wide-spread, sub-Gaussian centered distribution. It would model, e.g., in finance applications, returns of unknown levels but with known common shape. The diversity-preserving constraint could come from the necessity of putting all investments at a given round in a single asset while considering safe allocation policies that put, in expectation, a fraction of the capital in all assets.*

### 3.1 General regret lower bound

We restrict our attention to models satisfying the following technical assumption, which, as we discuss below, is mild and is satisfied, in particular, for convex models $\mathcal{D}$.

**Assumption 1.** *For all distributions $\nu \in \mathcal{D}$, for all real numbers $x$, if there exists $\zeta \in \mathcal{D}$ with expectation $\mathrm{E}(\zeta) > x$, then there exists $\zeta' \in \mathcal{D}$ such that $\mathrm{E}(\zeta') > x$ and $\mathrm{KL}(\nu, \zeta') < +\infty$.*

This assumption of course holds for models where all pairs of distributions exhibit finite Kullback-Leibler divergences (for instance, models given by exponential families). It also hold for convex models; indeed, for $\lambda \in (0, 1)$ small enough, $\zeta' = (1 - \lambda)\zeta + \lambda\nu$ belongs to $\mathcal{D}$ and still has an expectation larger than $x$, while the density of $\nu$ with respect to $\zeta'$ is upper bounded by $1/\lambda$, thus $\mathrm{KL}(\nu, \zeta') \leqslant 1/\lambda < +\infty$. Note that convex combinations of sub-Gaussian distributions with parameter $\sigma^2$ are also $\sigma^2$–sub-Gaussian, by convexity of the exponential function.

A somewhat typical restriction (also considered, e.g., by Lattimore & Szepesvári, 2017, Combes et al., 2017) will be issued concerning the bandit problems $\underline{\nu}$ possibly considered in the lower bounds: they should have a single optimal distribution in $\mathcal{P}$, which we denote by $\underline{p}^\star(\underline{\nu})$, i.e.,

$$\mathrm{Opt}(\underline{\nu}, \mathcal{P}) = \left\{\underline{p}^\star(\underline{\nu})\right\}.$$

Of course, when $\mathcal{P}$ is a polytope, that single optimal distribution $\underline{p}^\star(\underline{\nu})$ belongs to $\mathrm{Ext}(\mathcal{P})$.

The next, extremely mild, assumption is on arms: we require that there are no unnecessary arms given $\mathcal{P}$, i.e., that each arm in $[K]$ may be pulled for at least one distribution in $\mathcal{P}$, or equivalently if $\mathcal{P}$ is a polytope, in $\mathrm{Ext}(\mathcal{P})$.

**Assumption 2** (no unnecessary arms given $\mathcal{P}$). *For all $a \in [K]$, there exists $\underline{p} \in \mathcal{P}$ such that $p_a > 0$.*

Finally, given the regret upper bounds exhibited in Theorem 1, it is natural to restrict our attention to so-called uniformly fast convergent [UFC] strategies.

**Definition 5** (UFC strategies). *A strategy is uniformly fast convergent [UFC] over the model $\mathcal{D}$ given the diversity-preserving set $\mathcal{P}$ if for all bandit problems $\underline{\nu}$ in $\mathcal{D}$, its diversity-preserving regret satisfies $R_T = o(T^\alpha)$ for all $\alpha > 0$.*

The main result of this section is the following theorem, proved in Appendix B. We actually provide therein a general lower bound holding for large classes of models—mean-bounded and mean-unbounded ones.

**Theorem 2.** *Consider a mean-unbounded model $\mathcal{D}$ abiding by Assumption 1, and a diversity-preserving polytope $\mathcal{P}$ such that there are no unnecessary arms given $\mathcal{P}$ (see Definition 2). For all bandit problems $\underline{\nu}$ in $\mathcal{D}$ with a single optimal distribution in $\mathcal{P}$ denoted by $\underline{p}^\star(\underline{\nu})$, if*

$$\exists\, a \in [K] : p_a^\star(\underline{\nu}) = 0,$$

*then there exists $L_{\underline{\nu}} > 0$ such that for all strategies UFC over $\mathcal{D}$ given $\mathcal{P}$,*  $\qquad \liminf_{T \to \infty} \dfrac{R_T}{\ln T} \geqslant L_{\underline{\nu}}.$

*A closed-form expression of a suitable $L_{\underline{\nu}}$ is given by Lemma 4 in Appendix B.1.*

### 3.2 Characterization, or lack of characterization, of the optimal regret rates

Theorems 1 and 2 entail the following characterization of regret rates in the case of sub-Gaussian mean-unbounded models.

**Corollary 1.** *For a diversity-preserving polytope $\mathcal{P}$ such that there are no unnecessary arms given $\mathcal{P}$ (see Definition 2), for a mean-unbounded model $\mathcal{D}_{\sigma^2}$ composed of $\sigma^2$-sub-Gaussian distributions and abiding by Assumption 1, for all bandit problems $\underline{\nu}$ in $\mathcal{D}_{\sigma^2}$ with a single optimal distribution $\underline{p}^\star(\underline{\nu})$ in $\mathcal{P}$,*

- *if $\exists\, a \in [K] : p_a^\star(\underline{\nu}) > 0$, the diversity-preserving UCB strategy ensures that the regret $R_T$ is bounded;*
- *if $\exists\, a \in [K] : p_a^\star(\underline{\nu}) = 0$, then all strategies UFC over $\mathcal{D}$ given $\mathcal{P}$ suffer a logarithmic regret.*

On the contrary, the case of mean-bounded models is delicate. The (counter-)example below deals with the model of Bernoulli distributions. The characterization of rates obtained looks idiosyncratic: it seems intrinsically difficult to provide a general characterization of bounded regret by the expected payoffs in the case of mean-bounded models.

**Proposition 1.** *For the model $\mathcal{B}$ of all Bernoulli distributions, $K = 3$ arms, and some $\delta \in (0,1)$, consider the diversity-preserving segment $\mathcal{P}_\delta$ generated by $\underline{p}^{(1)} = (0, 1/2, 1/2)$ and $\underline{p}_\delta^{(2)} = (\delta, 0, 1-\delta)$. Let the bandit problem be $\underline{\nu} = \big(\mathrm{Ber}(0), \mathrm{Ber}(1/2), \mathrm{Ber}(0)\big)$. The setting introduced satisfies all assumptions of Theorem 2, except the mean-unboundedness of the model, and $\underline{p}^{(1)}$ is the unique optimal distribution for $\underline{\nu}$. Yet,*

- *if $\delta > 1/4$, there exists $L_{\underline{\nu}} > 0$ such that for all strategies UFC over $\mathcal{B}$ given $\mathcal{P}_\delta$, $\displaystyle\liminf_{T\to\infty} \frac{R_T}{\ln T} \geqslant L_{\underline{\nu}}$;*

- *if $\delta < 1/4$, a variant of the diversity-preserving UCB obtains $\displaystyle\limsup_{T\to\infty} R_T < +\infty$.*

## 4 Example of a diversity-preserving set $\mathcal{P}$ given by a ball

In this section, we discuss a specific example of a diversity-preserving set $\mathcal{P}$ that is curved with smooth boundary $\partial\mathcal{P}$. Our aim here is to illustrate that in the non-polytope case, there can be a wider variety of rates for the growth of regret—beyond the $\ln T$ rate and bounded-regret cases discussed earlier in the case of polytopes. Our conjecture (given the proof technique of Theorem 3 below) is that the $\ln^2 T$ rate for the regret exhibited for the specific example also holds more generally for convex sets with local quadratic curvatures. We leave the general study of the non-polytope case to future research.

**Specific example studied.** Our running example will be the intersection of the probability simplex with a Euclidean ball centered at the uniform distribution,

$$\mathcal{P}_r = \left\{ \underline{p} : \quad \sum_{a \in [K]} (p_a - 1/K)^2 \leqslant r^2 \right\}, \tag{3}$$

for $r$ small enough. This example differs from the case of polytopes in that the set of potential best actions becomes infinite (continuous): it is still given by extremal points, which are now given by the boundary $\partial\mathcal{P}_r$. In particular, what corresponds to the minimal gap $\Delta$ is now equal to 0.

In standard linear bandits with actions sets given by such convex sets $\mathcal{P}_r$, the logarithmic regret bounds, which depend on the inverse gap, become vacuous and the best problem-dependent rates degrade to $\sqrt{T}$, cf. Abbasi-Yadkori et al. (2011) and Lattimore & Szepesvári (2020). Banerjee et al. (2023, Theorem 3.3) proves that reaching sub-$\sqrt{T}$ regret on ellipsoids is essentially impossible by showing that any algorithm with worst-case regret smaller than $\sqrt{T}$ must explore all directions a proportion $\sqrt{T}$ of the time.

In the diversity-preserving setting, we state below that a version of the diversity-preserving UCB strategy enjoys a regret bound of order $\ln^2 T$ instead of the typical $\sqrt{T}$ bound that a linear bandit algorithm not taking into account the arms $A_t$ played would incur.

**Statement of the strategy.** The formulation of this diversity-preserving UCB strategy is almost identical to the Box B strategy, except that the set of extremal points is now given by the boundary $\partial\mathcal{P}_r$ of $\mathcal{P}_r$ and is infinite; we therefore specifically pick, in Step 1 of Box B:

$$\underline{p}_t \in \operatorname*{argmax}_{\underline{p} \in \partial\mathcal{P}_r} \big\langle \underline{p}, \underline{U}(t-1) \big\rangle. \tag{4}$$

Note in passing that computing $\underline{p}_t$ comes at no computational cost: a simple closed-form expression follows from Lemma 6 of Appendix D.

If the radius $r$ is small enough, then $\mathcal{P}_r$ is contained in the relative interior of the probability simplex, guaranteeing that every pure action $a \in [K]$ gets some minimal positive chance to be played at any round.

More precisely, the condition reads

$$r < r_{\lim} := \frac{1}{K}\sqrt{1 + \frac{1}{K-1}}, \qquad \text{for which} \qquad \min_{\underline{p} \in \mathcal{P}_r} \min_{a \in [K]} p_a \geqslant r_{\lim} - r\,. \tag{5}$$

The value $r_{\lim}$ is the distance from the uniform distribution vector to the uniform distribution on all but one pure actions. The following asymptotic result is proved in Appendix D.

**Theorem 3.** *Assume that $r < r_{\lim}$. Consider a sub-Gaussian model $\mathcal{D}$ with parameter $\sigma^2$, known and used by the diversity-preserving UCB strategy (4) on $\mathcal{P}_r$. For any bandit problem $\underline{\nu}$ in $\mathcal{D}$, the regret of this strategy is at most squared-logarithmic:*

$$\limsup_{T \to \infty} \frac{R_T}{\ln^2 T} < +\infty\,.$$

While this result should generalize to other strongly convex diversity-preserving sets, we stick to the representative example of $\mathcal{P}_r$, which simplifies the exposition of the proof. In particular, we use the simple shape of $\mathcal{P}_r$ to provide explicit formulas for the optimal distribution(s) and optimal payoff (see Lemmas 6 and 7 in Appendix D). Similarly, we focused on the asymptotic $\ln^2 T$ rate with no attempt to provide a closed-form regret bound with explicit constants. These constants would depend on $\underline{\mu}$ and on the curvature of $\mathcal{P}$ in a complex way, and we chose to ignore this aspect here.

## 5 Some experiments on synthetic data

In this final section, we perform some (simple and preliminary) experiments that merely illustrate the dual behavior of the regret: either bounded or growing at a $\ln T$ rate. We believe that a more extensive empirical comparison would be interesting but would be out of the scope we targeted for this article.

**Experimental setting.** We consider $K = 3$ arms and the model $\mathcal{D}$ of Bernoulli distributions. The diversity-preserving set $\mathcal{P}$ is the triangle generated by

$$\underline{p}^{(1)} = (0,\, 0.2,\, 0.8)\,, \qquad \underline{p}^{(2)} = (0.6,\, 0.2,\, 0.2)\,, \qquad \text{and} \qquad \underline{p}^{(3)} = (0,\, 0.8,\, 0.2)\,.$$

We consider the bandit problems $\underline{\nu}_\alpha$ with expectations

$$\underline{\mu}_\alpha = (1/2 - \alpha,\, 1/3,\, 1/2 + \alpha)\,, \qquad \text{where} \qquad \alpha \in \{-0.1,\, 0.1\}\,.$$

**Lemma 1.** *For $\alpha = -0.1$, the unique optimal distribution is $\underline{p}^{(2)}$, which puts a positive probability mass on all actions. For $\alpha = 0.1$, the unique optimal distribution is $\underline{p}^{(1)}$, which does not put any probability mass on action 1.*

*Proof.* First, the distribution $\underline{p}^{(3)}$ is always dominated either by $\underline{p}^{(1)}$ or by $\underline{p}^{(2)}$, and is therefore never an optimal distribution: for all $\alpha \in \{-0.1,\, 0.1\}$,

$$\langle \underline{p}^{(3)} - \underline{p}^{(1)},\, \underline{\mu}_\alpha \rangle = 0.6\left(1/3 - (1/2 + \alpha)\right) = -(1 + 0.6\alpha) < 0\,;$$
$$\langle \underline{p}^{(3} - \underline{p}^{(2)},\, \underline{\mu}_\alpha \rangle = 0.6\left(-(1/2 - \alpha) + 1/3\right) = -1 + 0.6\alpha < 0\,.$$

We now compare the mixed actions $\underline{p}^{(1)}$ and $\underline{p}^{(2)}$:

$$\langle \underline{p}^{(1)} - \underline{p}^{(2)},\, \underline{\mu}_\alpha \rangle = 0.6\left(-(1/2 - \alpha) + (1/2 + \alpha)\right) = 1.2\alpha\,.$$

The sign of $\alpha$ thus determines which of $\underline{p}^{(1)}$ or $\underline{p}^{(2)}$ is the optimal distribution. $\qquad\square$

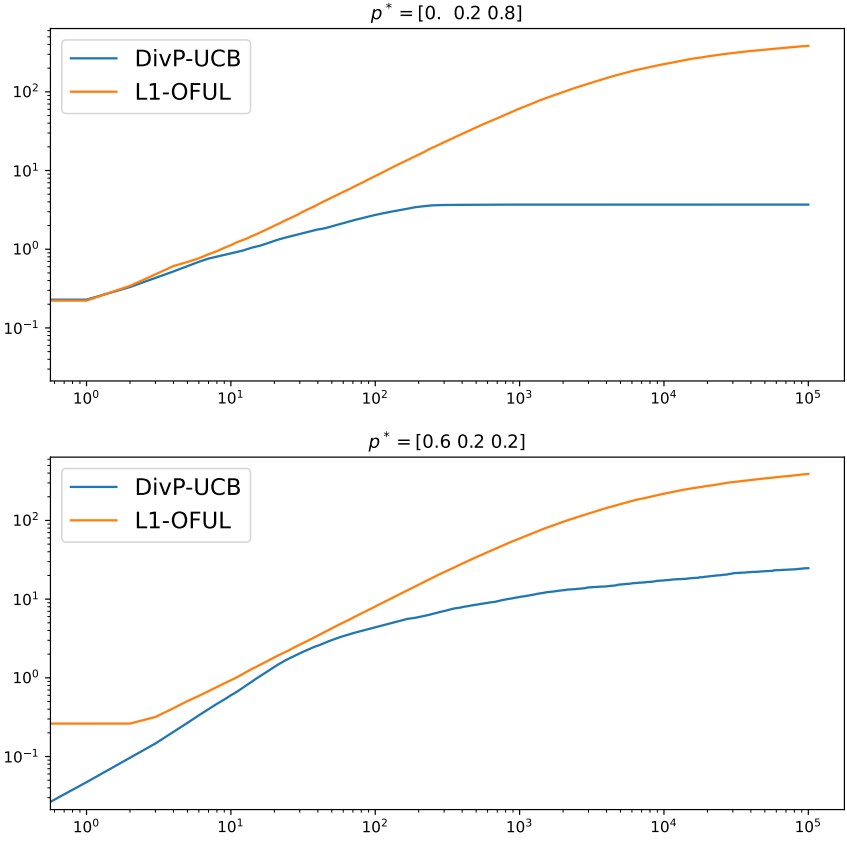

Figure 1: Estimated expected cumulative regret over time, in the case $\alpha = -0.1$ [*top figure*, bounded regret] and $\alpha = 0.1$ [*bottom figure*, $\ln T$ growth], for the two algorithms considered. Solid lines report empirical means while shaded areas correspond to $\pm 2$ standard errors of the series defining the empirical means.

**Numerical experiments.** We ran the diversity-preserving UCB algorithm (see Box B, abbreviated DivP-UCB below), as well as Algorithm 2 (Constrained-$L_1$-OFUL, abbreviated $L_1$-OFUL below) of Celis et al. (2019). We did so on each of the two problems $\underline{\nu}_\alpha$, over $T = 100{,}000$ time steps, for $N = 100$ runs. The expected regret suffered by each algorithm is estimated by the empirical averages of pseudo-regrets observed on the $N$ runs:

$$\widehat{R}_T(\underline{\nu}_\alpha) = \frac{1}{N} \sum_{i=1}^{N} \widehat{R}_T(\underline{\nu}_\alpha, i), \quad \text{where} \quad \widehat{R}_T(\underline{\nu}_\alpha, i) = \sum_{t=1}^{T} \left\langle \underline{p}^\star(\underline{\nu}_\alpha) - \underline{p}_t(\alpha, i), \, \underline{\mu}_\alpha \right\rangle,$$

and where we denoted by $\underline{p}_t(\alpha, i)$ the mixed action chosen at round $t$, during the $i$–th run, for problem $\underline{\nu}_\alpha$.

Figures 1 report the estimates $R_T(\underline{\nu}_\alpha)$ obtained (solid lines); shaded areas correspond to $\pm 2$ standard errors of the series $R_T(\underline{\nu}_\alpha, i)$ used in the definition of the $R_T(\underline{\nu}_\alpha)$ as empirical averages.

As expected by combining Theorem 1 and Lemma 1, we observe a logarithmic regret growth when $\alpha = -0.1$ and bounded regret when $\alpha = 0.1$ for DivP-UCB, whereas L1-OFUL incurs logarithmic regret in both cases. It also turns out that on this specific example, DivP-UCB performs better than L1-OFUL. (Note also that L1-OFUL is computationally more costly than DivP-UCB, since it needs to solve $2K$ linear programs at every time step, instead of a single one for DivP-UCB.)

## 6 Conclusion and limitations

We revisited the diversity-preserving variant of $K$–armed bandits introduced by Celis et al. (2019): we considered the same framework as introduced therein (while extending it to sub-Gaussian models), with diversity-preserving sets $\mathcal{P}$ given by polytopes; but we stated and discussed a more satisfactory UCB-based strategy, controlling the regret by a $\ln T$ rate in general and even achieving a bounded regret in the case where all arms get a positive probability mass by all optimal distributions—i.e., when diversity is desirable. We showed that this dual behavior (and the associated condition) were optimal for mean-unbounded models, while pointing out, through a Bernoulli counter-example, the intrinsic difficulties in characterizing the rates in the case of mean-bounded models. The latter issue is the first question left for future research. We then also discussed a specific example of a diversity-preserving set $\mathcal{P}$ not given by a polytope, where the regret grows at worst at a $\ln^2 T$ rate. Here as well, building a general theory for curved diversity-preserving sets $\mathcal{P}$ is left for future research. Finally, we proposed some simple and preliminary numerical simulations, that could be extended.

### Acknowledgments

The work of Sébastien Gerchinovitz and Jean-Michel Loubes has benefitted from the AI Interdisciplinary Institute ANITI, which is funded by the French "Investing for the Future – PIA3" program under the Grant agreement ANR-19-PI3A-0004. Sébastien Gerchinovitz gratefully acknowledges the support of the DEEL project (`https://www.deel.ai/`).

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

# A    Proof of Theorem 1: Analysis of the diversity-preserving UCB policy

The $\ln T$ bound of Theorem 1 is proved in Appendix A.1 by showing, in view of (2), that suboptimal distributions $\underline{p} \in \mathrm{Ext}(\mathcal{P})$ are unlikely to be played more than $\ln T$ times. The analysis mimics and adapts the proof scheme corresponding to UCB run on $\mathrm{Ext}(\mathcal{P})$, with three new ingredients specifically underlined.

The proof of the constant regret bound of Theorem 1 may be found in Appendix A.2 and follows a completely different logic. We first show that optimal distributions are typically played at least half of the time. This entails, because $p^{\star}_{\min}(\underline{\nu}) > 0$, that each pure action $a \in [K]$ is played linearly many times. Therefore, all estimates are sharp, and little regret is suffered.

## A.1    Proof of the $\ln T$ bound in Theorem 1

We want to control the $\mathbb{E}\big[N_{\underline{p}}(t)\big]$ by $\ln T$, however, the favorable events at round $t \geqslant 1$ hold rather for quantities based on how often the pure actions were pulled:

$$\mathcal{E}(t) = \left\{ \forall a \in [K], \quad \big|\mu_a - \widehat{\mu}_a(t)\big| \leqslant \sqrt{\frac{8\sigma^2 \ln t}{\max\big\{N_a(t), 1\big\}}} \right\} \quad \text{and} \quad \mathcal{E}'(t) = \left\{ \forall a \in [K], \quad \sqrt{\frac{8\sigma^2 \ln t}{\max\big\{N_a(t), 1\big\}}} < \frac{\Delta}{2} \right\}$$

We also introduce the following events, for any $\underline{p} \in \mathcal{P}$, though we will use them only for $\underline{p} \in \mathrm{Ext}(\mathcal{P}) \backslash \mathrm{Opt}(\underline{\nu}, \mathcal{P})$ in the sequel:

$$\mathcal{E}''(\underline{p}, t) = \left\{ \sum_{a=1}^{K} p_a \sqrt{\frac{8\sigma^2 \ln t}{\max\big\{N_a(t), 1\big\}}} < \frac{\Delta}{2} \right\}.$$

A first new ingredient consists of the following inequalities, obtained by distinguishing whether $p_a = 0$ or $p_a > 0$ and by Jensen's equality for the square root: for all $\underline{p} \in \mathcal{P}$, all $t \leqslant T$, and all $n \geqslant (65K\sigma^2/\Delta^2) \ln T$,

$$2\sqrt{8\sigma^2 \ln t} \sum_{a \in [K]} p_a \frac{1}{\sqrt{\max\big\{np_a/2, 1\big\}}} \leqslant \frac{8\sqrt{\sigma^2 \ln t}}{\sqrt{n}} \sum_{a \in [K]} \sqrt{p_a} \leqslant \frac{8\sqrt{\sigma^2 \ln t}}{\sqrt{n}} \sqrt{K} < \Delta,$$

thus the following inclusion:

$$\bigcap_{a \in [K]} \big\{N_a(t) \geqslant np_a/2\big\} = \bigcap_{a:p_a>0} \big\{N_a(t) \geqslant np_a/2\big\} \subseteq \mathcal{E}''(\underline{p}, t). \tag{6}$$

Note also the inclusion $\mathcal{E}'(t) \subseteq \mathcal{E}''(\underline{p}, t)$, valid for all $\underline{p} \in \mathcal{P}$.

Now, the second new ingredient, consisting of the lemma below, is the key to relate the numbers of times $N_{\underline{p}}(t)$ a suboptimal distribution $\underline{p} \in \mathrm{Ext}(\mathcal{P})$ is picked to the numbers of draws $N_a(t)$ of pure actions $a \in [K]$.

**Lemma 2.** *Fix $\underline{p} \in \mathrm{Ext}(\mathcal{P})$, and denote by $p_{\min>0} = \min\big\{p_a : a \in [K] \text{ s.t. } p_a > 0\big\} > 0$ its minimal positive component. Then, for all $t \geqslant 1$, all $n \geqslant (10/p_{\min>0}) \ln T$, and all $a \in [K]$,*

$$\mathbb{P}\Big(\big\{N_{\underline{p}}(t) \geqslant n\big\} \cap \big\{N_a(t) < np_a/2\big\}\Big) \leqslant \frac{1}{T}.$$

*Proof.* We only need to show the inequality for $a \in [K]$ such that $p_a > 0$. We note that

$$N_a(t) \geqslant \sum_{s=1}^{t} \mathbb{1}_{\{\underline{p}_s = \underline{p}\}} \, \mathbb{1}_{\{A_s = a\}}; \tag{7}$$

thus, by optional skipping[2] (see Theorem 5.2 of Doob, 1953, Chapter III, p. 145, see also Chow & Teicher, 1988, Section 5.3), the distribution of $N_a(t)$ on the event $\big\{N_{\underline{p}}(t) \geqslant n\big\}$ is larger than the distribution of a

---

[2]Sometimes called optional sampling.

random variable $B_{n,a}$ with binomial distribution of parameters $n$ and $p_a$. In particular,

$$\mathbb{P}\Big(\big\{N_{\underline{p}}(t) \geqslant n\big\} \cap \big\{N_a(t) < np_a/2\big\}\Big)$$

$$\leqslant \mathbb{P}\big(B_{n,a} < np_a/2\big) = \mathbb{P}\big(B_{n,a} - np_a < -np_a/2\big) \leqslant \exp\left(-\frac{\varepsilon^2}{2(v + b\varepsilon/3)}\right) \leqslant \exp\left(-\frac{np_a}{8(1 + 1/6)}\right),$$

where, for the final inequality, we applied Bernstein's inequality (see, e.g., Boucheron et al., 2013, end of Section 2.7, Equation 2.10) with variance $v = n\,p_a(1 - p_a)$, upper bound $b = 1$ on the range, and deviation $\varepsilon = np_a/2$. Substituting the bound on $n$ concludes the proof. $\qquad\square$

The rest of the analysis is essentially standard. The aim is to control each of the following expectations, for $\underline{p} \in \mathrm{Ext}(\mathcal{P}) \setminus \mathrm{Opt}(\underline{\nu}, \mathcal{P})$ and where $n_{\underline{p}} \geqslant 1$ is defined later:

$$\mathbb{E}\big[N_{\underline{p}}(T)\big] \leqslant n_{\underline{p}} + \sum_{t=n_{\underline{p}}}^{T-1} \mathbb{P}\big\{\underline{p}_{t+1} = \underline{p} \ \text{ and } \ N_{\underline{p}}(t) \geqslant n_{\underline{p}}\big\}. \tag{8}$$

We first note that for $t \geqslant 1$, for all $\underline{p} \in \mathrm{Ext}(\mathcal{P}) \setminus \mathrm{Opt}(\underline{\nu}, \mathcal{P})$,

$$\big\{\underline{p}_{t+1} = \underline{p}\big\} \subseteq \overline{\mathcal{E}(t)} \cup \overline{\mathcal{E}''(\underline{p}, t)} \subseteq \overline{\mathcal{E}(t)} \cup \overline{\mathcal{E}'(t)}; \tag{9}$$

indeed, on $\mathcal{E}(t) \cap \mathcal{E}''(\underline{p}, t)$, for $\underline{p}^\star \in \mathrm{Opt}(\underline{\nu}, \mathcal{P})$, by definitions of these sets and of $\underline{U}(t)$,

$$\langle \underline{p}, \underline{U}(t) \rangle = \langle \underline{p}, \widehat{\mu}(t) \rangle + \sum_{a=1}^K p_a \sqrt{\frac{8\sigma^2 \ln t}{\max\{N_a(t), 1\}}} \leqslant \langle \underline{p}, \underline{\mu} \rangle + 2\sum_{a=1}^K p_a \sqrt{\frac{8\sigma^2 \ln t}{\max\{N_a(t), 1\}}}$$

$$< \langle \underline{p}, \underline{\mu} \rangle + \Delta \leqslant \langle \underline{p}, \underline{\mu} \rangle + \Delta(\underline{p}) = \langle \underline{p}^\star, \underline{\mu} \rangle \leqslant \langle \underline{p}^\star, \widehat{\mu}(t) \rangle + \sum_{a=1}^K p_a^\star \sqrt{\frac{8\sigma^2 \ln t}{\max\{N_a(t), 1\}}} = \langle \underline{p}^\star, \underline{U}(t) \rangle,$$

while $\big\{\underline{p}_{t+1} = \underline{p}\big\}$ requires $\langle \underline{p}, \underline{U}(t) \rangle \geqslant \langle \underline{p}^\star, \underline{U}(t) \rangle$. Let

$$n_{\underline{p}} = \max\left\{\frac{65K}{\Delta^2} \ln T, \ \frac{10}{p_{\min > 0}} \ln T, \ 1 + \frac{1}{8\sigma^2} \max_{a \in [K]} (\mu_a - u_0)^2\right\}; \tag{10}$$

the third element in the maximum will turn useful in the application of Lemma 3 below. For each distribution $\underline{p} \in \mathrm{Ext}(\mathcal{P}) \setminus \mathrm{Opt}(\underline{\nu}, \mathcal{P})$, the inclusions (9) and then (6) entail

$$\big\{\underline{p}_{t+1} = \underline{p}\big\} \cap \big\{N_{\underline{p}}(t) \geqslant n_{\underline{p}}\big\} \subseteq \overline{\mathcal{E}(t)} \cup \Big(\overline{\mathcal{E}''(\underline{p}, t)} \cap \big\{N_{\underline{p}}(t) \geqslant n_{\underline{p}}\big\}\Big) \subseteq \overline{\mathcal{E}(t)} \cup \bigcup_{a \in [K]} \big\{N_{\underline{p}}(t) \geqslant n_{\underline{p}}\big\} \cap \big\{N_a(t) < n_{\underline{p}} p_a/2\big\}.$$

Substituting this bound into (8), resorting to unions bounds and to Lemma 2, yields

$$\mathbb{E}\big[N_{\underline{p}}(T)\big] \leqslant n_{\underline{p}} + K + \sum_{t=n_{\underline{p}}}^{T-1} \mathbb{P}\big(\overline{\mathcal{E}(t)}\big) \leqslant n_{\underline{p}} + K + K \sum_{t=n_{\underline{p}}}^{T-1} (2t\,t^{-4}) \leqslant n_{\underline{p}} + 2K, \tag{11}$$

where we applied Lemma 3 below for each $a \in [K]$ and with $\delta = t^{-4}$, which satisfies the condition required therein given that $t \geqslant n_{\underline{p}}$. The proof is concluded by resorting to the decomposition (2), to obtain

$$R_T \leqslant \sum_{\underline{p} \in \mathrm{Ext}(\mathcal{P}) \setminus \mathrm{Opt}(\underline{\nu}, \mathcal{P})} \Delta_{\underline{p}}(n_{\underline{p}} + 2K), \tag{12}$$

which is of the claimed form $C_{\underline{\nu}} \ln T + c_{\underline{\nu}}$. In the derivation of this regret bound, we targeted simplicity and did not try to improve the constants $C_{\underline{\nu}}$ and $c_{\underline{\nu}}$.

Lemma 3 is an essentially standard concentration result for stochastic bandits; the only adaptation therein (the third new ingredient) is handling the case where $N_a(t) = 0$.

**Lemma 3.** *Consider a model $\mathcal{D}_{\sigma^2}$ with $\sigma^2$–sub-Gaussian distributions, and fix a bandit problem $\underline{\nu}$ in $\mathcal{D}_{\sigma^2}$. For $t \geqslant 1$, if the actions $A_1, \ldots, A_t$ and rewards $Y_1, \ldots, Y_t$ were generated according to the protocol of Box A, then, for all $a \in [K]$, for all $\delta > 0$ with $2\ln(1/\delta) > (\mu_a - u_0)^2/\sigma^2$,*

$$\mathbb{P}\left\{ \left|\mu_a - \widehat{\mu}_a(t)\right| \geqslant \sqrt{\frac{2\sigma^2 \ln(1/\delta)}{\max\{N_a(t), 1\}}} \right\} \leqslant 2t\delta\,.$$

*Proof.* Again by optional skipping (see the proof of Lemma 2), by denoting by $\widehat{\mu}_{a,n}$ an empirical average of $n \geqslant 1$ i.i.d. random variables with distribution $\nu_a$, and by using the convention $\widehat{\mu}_{a,0} = u_0$, we have

$$\mathbb{P}\left\{ \left|\mu_a - \widehat{\mu}_a(t)\right| \geqslant \sqrt{\frac{2\sigma^2 \ln(1/\delta)}{\max\{N_a(t), 1\}}} \right\} \leqslant \mathbb{P}\left\{ \exists n \in \{0, 1, \ldots, t\} \, : \, \left|\mu_a - \widehat{\mu}_{a,n}\right| \geqslant \sqrt{\frac{2\sigma^2 \ln(1/\delta)}{\max\{n, 1\}}} \right\}$$

$$\leqslant 0 + \sum_{n=1}^{t} \mathbb{P}\left\{ \left|\mu_a - \widehat{\mu}_{a,n}\right| \geqslant \sqrt{\frac{2\sigma^2 \ln(1/\delta)}{n}} \right\} \leqslant \sum_{n=1}^{t} 2\delta = 2t\delta\,,$$

where the case $n = 0$ was dropped in the union bound because $|\mu_a - u_0| < \sqrt{2\sigma^2 \ln(1/\delta)}$ by assumption, and where the final inequalities follow from the Cramér–Chernoff inequality (see, e.g., Lattimore & Szepesvári, 2020, Corollary 5.1). $\qquad\square$

### A.2 Proof of the constant regret bound in Theorem 1

As indicated at the beginning of Appendix A, the proof of the constant regret bound of Theorem 1 follows a completely different logic. For instance, in Appendix A.1, the sets $\mathcal{E}'(t)$ were instrumental in the proof but we had not controlled their probabilities—which constitutes the core of the analysis here. To do so, we show that optimal distributions are typically played at least half of the time; this is the main contribution of this proof. Then, because $p^\star_{\min}(\underline{\nu}) > 0$, we know that each pure action $a \in [K]$ is played linearly many times, which cannot happen on the events $\overline{\mathcal{E}'(t)}$, where at least one action is only played logarithmically many times. This proof strategy for bounded regret was used in Lattimore & Munos (2014). We face the additional technical challenge here that we do not control with certainty the number of pulls of every pure arm because of the randomness in generating the $A_t$ from the $p_t$; we handle this by carefully applying Bernstein's inequality.

**Step 1: Preparation.** We fix a threshold $t_0 \geqslant 8 + \max_{a \in [K]} (\mu_a - u_0)^2/(8\sigma^2)$ such that

$$\forall t \geqslant t_0, \qquad \frac{t}{2} p^\star_{\min}(\underline{\nu}) - \frac{32\sigma^2 \ln t}{\Delta^2} \geqslant \sqrt{t \ln t} \qquad \text{and} \qquad \frac{\Delta t}{4} - \sqrt{8\sigma^2 \ln t}\left(1 + 2\sqrt{t-1}\right) > \sqrt{8\sigma^2 t \ln^2 t}\,. \tag{13}$$

For example, with the convention that the $\ln \ln x = -\infty$ if $x \leqslant 1$, the constraints above are satisfied with the threshold $t_0$ such that

$$\ln t_0 = \max\left\{ 2 + \frac{1}{8\sigma^2}, \; \ln \frac{\sigma^2}{\Delta^2 \, p^\star_{\min}(\underline{\nu})^2} + 3\ln\ln \frac{18\,432\,\sigma^2}{\Delta^2 \, p^\star_{\min}(\underline{\nu})^2} + 10 \right\}. \tag{14}$$

(*Note to reviewers: for the sake of concision, we decided to omit the half-page of calculations that lead to this bound. We could of course add it if deemed necessary.*)

By (9), we first note that

$$R_T \leqslant R_{t_0} + \max_{a \in [K]} \mu_a \sum_{t=t_0}^{T-1} \left( \mathbb{P}\left(\overline{\mathcal{E}(t)}\right) + \mathbb{P}\left(\overline{\mathcal{E}'(t)}\right) \right) \leqslant R_{t_0} + \max_{a \in [K]} \mu_a \left( K + \sum_{t=t_0}^{T-1} \mathbb{P}\left(\overline{\mathcal{E}'(t)}\right) \right),$$

where the final inequality follows from a bound proved in (11), given the first condition on $t_0$. The key step is the decomposition

$$\overline{\mathcal{E}'(t)} \subseteq \left\{ N_\star(t) < t/2 \right\} \cup \left( \overline{\mathcal{E}'(t)} \cap \left\{ N_\star(t) \geqslant t/2 \right\} \right), \qquad \text{where} \qquad N_\star(t) = \sum_{s=1}^{t} \sum_{\underline{p} \in \mathrm{Opt}(\underline{\nu}, \mathcal{P})} \mathbb{1}_{\{\underline{p}_s = \underline{p}\}}$$

denotes the number of times optimal distributions are played.

**Step 2 (core step): Optimal distributions are typically played at least half of the time.** We first deal with the events $\{N_\star(t) < t/2\}$, where $t \geqslant t_0$. I.e., contrary in the proof of Appendix A.1, we do not only control $N_\star(t)$ in expectation, but in high probability. To that end, we consider first

$$\{N_\star(t) < t/2\} \cap \mathcal{E}(\lfloor t/4 \rfloor : \infty), \qquad \text{where} \quad \mathcal{E}(\lfloor t/4 \rfloor : \infty) = \bigcap_{s=\lfloor t/4 \rfloor}^{\infty} \mathcal{E}(s).$$

By a classic UCB argument, for all $s \geqslant 1$, by definition of $\mathcal{E}(s)$,

$$\text{on } \mathcal{E}(s), \qquad 0 \leqslant U_a(s) - \mu_a = \widehat{\mu}_a(s) + \sqrt{\frac{8\sigma^2 \ln s}{\max\{N_a(s), 1\}}} - \mu_a \leqslant \text{UCB}_{s,a} \stackrel{\text{def}}{=} 2\sqrt{\frac{8\sigma^2 \ln s}{\max\{N_a(s), 1\}}},$$

thus, for any optimal $p^\star \in \text{Opt}(\nu, \mathcal{P})$, using also the definition of $\underline{p}_{s+1}$ as some empirical best distribution, we have, on $\mathcal{E}(s)$,

$$\langle \underline{p}^\star - \underline{p}_{s+1}, \underline{\mu} \rangle = \underbrace{\langle \underline{p}^\star, \underline{\mu} - \underline{U}(s) \rangle}_{\leqslant 0} + \underbrace{\langle \underline{p}^\star - \underline{p}_{s+1}, \underline{U}(s) \rangle}_{\leqslant 0} + \langle \underline{p}_{t+1}, \underbrace{\underline{U}(s) - \underline{\mu}}_{\leqslant (\text{UCB}_{s,a})_{a \in [K]}} \rangle \leqslant 2 \sum_{a \in [K]} p_{s+1,a} \sqrt{\frac{8\sigma^2 \ln s}{\max\{N_a(s), 1\}}}.$$

This inequality and the definition of $\Delta$ as the smallest gap over $\text{Ext}(\mathcal{P})$ yield, on $\mathcal{E}(\lfloor t/4 \rfloor : \infty)$,

$$\sum_{s=\lfloor t/4 \rfloor}^{t} \sum_{a \in [K]} p_{s,a} \sqrt{\frac{8\sigma^2 \ln(s-1)}{\max\{N_a(s-1), 1\}}} \geqslant \sum_{s=\lfloor t/4 \rfloor}^{t} \langle \underline{p}^\star - \underline{p}_s, \underline{\mu} \rangle \geqslant \Delta \big( t + 1 - \lfloor t/4 \rfloor - N_\star(t) \big).$$

As $t \geqslant t_0 \geqslant 8$ and as the sums are over non-negative terms, we proved so far, on $\mathcal{E}(\lfloor t/4 \rfloor : \infty)$:

$$\sum_{s=2}^{t} \sum_{a \in [K]} p_{s,a} \sqrt{\frac{8\sigma^2 \ln(s-1)}{\max\{N_a(s-1), 1\}}} \geqslant \Delta \big( t + 1 - \lfloor t/4 \rfloor - N_\star(t) \big). \tag{15}$$

We introduce the martingale

$$M_{\Sigma, t} \stackrel{\text{def}}{=} \sum_{s=2}^{t} \sum_{a \in [K]} \big( p_{s,a} - \mathbb{1}_{\{A_s = a\}} \big) \sqrt{\frac{8\sigma^2 \ln(s-1)}{\max\{N_a(s-1), 1\}}}.$$

It turns out that for each $a \in [K]$, as $N_a(s-1)$ increases by 1 if and only if $A_s = a$, and otherwise remains unchanged, we have the crude deterministic bound:

$$\sum_{s=2}^{t} \mathbb{1}_{\{A_s = a\}} \sqrt{\frac{8\sigma^2 \ln(s-1)}{\max\{N_a(s-1), 1\}}} \leqslant \sqrt{8\sigma^2 \ln t} \sum_{s=2}^{t} \frac{\mathbb{1}_{\{A_s = a\}}}{\sqrt{\max\{N_a(s-1), 1\}}}$$

$$\leqslant \sqrt{8\sigma^2 \ln t} \sum_{n=0}^{t-1} \frac{1}{\sqrt{\max\{n, 1\}}} \leqslant \sqrt{8\sigma^2 \ln t} \left( 1 + 2\sqrt{t-1} \right),$$

where we used that $N_a(t-1) \leqslant t-1$ to determine the range of values for $n$ in the sum. The inequality above, together with the relationship (15) and the definition of $M_{\Sigma, t}$, entails that

$$\{N_\star(t) < t/2\} \cap \mathcal{E}(\lfloor t/4 \rfloor : \infty) \subseteq \left\{ M_{\Sigma, t} + \sqrt{8\sigma^2 \ln t} \big( 1 + 2\sqrt{t-1} \big) > \Delta \big( t + 1 - \lfloor t/4 \rfloor - t/2 \big) \right\}$$

$$\subseteq \left\{ M_{\Sigma, t} > \sqrt{8\sigma^2 t \ln^2 t} \right\}, \tag{16}$$

where the second inclusion comes from the final condition 13 on $t_0$. As indicated later, the probability of the right-hand side is smaller than $1/t^2$.

**Step 3: The remaining events.** We now turn to the events $\overline{\mathcal{E}'(t)} \cap \{N_\star(t) \geqslant t/2\}$, and show that they are unlikely; the intuition is that when $N_\star(t)$ is linearly large, because of the condition $p^\star_{\min}(\underline{\nu}) > 0$, the $N_a(t)$ should also be linearly large. More precisely, by the respective definitions of $\overline{\mathcal{E}'(t)}$ and of $p^\star_{\min}(\underline{\nu}) > 0$,

$$\overline{\mathcal{E}'(t)} = \bigcup_{a \in [K]} \left\{ N_a(t) \leqslant \frac{32\sigma^2 \ln t}{\Delta^2} \right\} \quad \text{and} \quad \forall a \in [K], \quad \sum_{s=1}^t p_{s,a} \geqslant \sum_{s=1}^t \mathbb{1}_{\{\underline{p}_s \in \mathrm{Opt}(\underline{\nu}, \mathcal{P})\}} \, p^\star_{\min}(\underline{\nu}) = N_\star(t) \, p^\star_{\min}(\underline{\nu}) \,.$$

We introduce the martingales $M_{a,t} = N_a(t) - \sum_{s=1}^t p_{s,a}$ and get, for $t \geqslant t_0$,

$$\overline{\mathcal{E}'(t)} \cap \{N_\star(t) \geqslant t/2\} \subseteq \bigcup_{a \in [K]} \left\{ M_{a,t} \leqslant \frac{32\sigma^2 \ln t}{\Delta^2} - \frac{t}{2} \, p^\star_{\min}(\underline{\nu}) \right\} \subseteq \bigcup_{a \in [K]} \left\{ M_{a,t} \leqslant -\sqrt{t \ln t} \right\}, \tag{17}$$

where the final inequality is by the conditions (13) on $t_0$. We bound below the probability of the right-hand side.

**Step 4: Taking probabilities, via the Hoeffding–Azuma inequality.** Collecting the bounds (16) and (17) together, and applying union bounds, we proved so far

$$\sum_{t=t_0}^{T-1} \mathbb{P}\left(\overline{\mathcal{E}'(t)}\right) \leqslant \sum_{t=t_0}^{T-1} \mathbb{P}\left\{M_{\Sigma,t} > \sqrt{8\sigma^2 t \ln^2 t}\right\} + \sum_{t=t_0}^{T-1} \mathbb{P}\left(\overline{\mathcal{E}\big(\lfloor t/4 \rfloor : \infty\big)}\right) + \sum_{a \in [K]} \sum_{t=t_0}^{T-1} \mathbb{P}\left\{M_{a,t} \leqslant -\sqrt{t \ln t}\right\},$$

and now show that each of these sums is finite. We note that $\mathbb{P}\left(\overline{\mathcal{E}(t)}\right) \leqslant 2Kt^{-3}$ for $t \geqslant t_0$, as already shown in (11), thus, by union bounds,

$$\sum_{t=t_0}^{T-1} \mathbb{P}\left(\overline{\mathcal{E}\big(\lfloor t/4 \rfloor : \infty\big)}\right) \leqslant \sum_{t=t_0}^{T-1} \sum_{s \geqslant \lfloor t/4 \rfloor} 2Ks^{-3} \leqslant \sum_{t=t_0}^{T-1} \frac{K}{(t/4-1)^2} \leqslant \sum_{t=8}^{\infty} \frac{4K}{(t-4)^2} \leqslant 2K \,.$$

For each $a \in [K]$, since the $\mathbb{1}_{\{A_s = a\}} - p_{s,a}$ form martingale increments with values in a predictable range of total width 1, we have, by the Hoeffding–Azuma inequality, that for all $t \geqslant 1$ and all $\varepsilon > 0$,

$$\mathbb{P}\{M_{a,t} \leqslant -\varepsilon\} = \mathbb{P}\left\{N_a(t) - \sum_{s=1}^t p_{s,a} \leqslant -\varepsilon\right\} \leqslant \exp\left(-2\varepsilon^2/t\right) \qquad \text{thus} \qquad \mathbb{P}\left\{M_{a,t} \leqslant -\sqrt{t \ln t}\right\} \leqslant \frac{1}{t^2} \,. \tag{18}$$

Similarly, the $M_{\Sigma,t}$ are sums of $t-1$ martingale increments with values in predictable range of total widths each smaller than $\sqrt{8\sigma^2 \ln t}$, so that, again by the Hoeffding–Azuma inequality,

$$\mathbb{P}\left\{M_{\Sigma,t} \geqslant \varepsilon\sqrt{8\sigma^2 \ln t}\right\} \leqslant \exp\left(-2\varepsilon^2/(t-1)\right) \qquad \text{thus} \qquad \mathbb{P}\left\{M_{\Sigma,t} > \sqrt{8\sigma^2 \ln t}\sqrt{t \ln t}\right\} \leqslant \frac{1}{t^2} \,.$$

Collecting all bounds and recalling that $t_0 \geqslant 8$, we proved the following closed-form finite regret bound, where we recall that $t_0$ was defined in (13) and where $R_{t_0}$ can be bounded by the general $\ln t_0$ closed-form regret bound (12) proved above:

$$\sum_{t=t_0}^{T-1} \mathbb{P}\left(\overline{\mathcal{E}'(t)}\right) \leqslant 3K + 1 \,, \qquad \text{thus} \qquad R_T \leqslant R_{t_0} + \max_{a \in [K]} \mu_a (4K + 1) \,. \tag{19}$$

Combine the logarithmic regret bounds (10) and (12) with the upper bound (14) on $\ln t_0$ to get a closed-form expression of the final regret.

# B  Proofs of Theorem 2: Lower bound on the diversity-preserving regret

The main difference with respect to the classic proof schemes for lower bounds (Lai & Robbins, 1985, Graves & Lai, 1997, Garivier et al., 2019) is described in Remark 1: eventually, arms $A_t$ are played, so that the information gain should be quantified in terms of the $A_t$ and is larger than if it was quantified in terms of the distributions $\underline{p}_t$ used; yet, the UFC constraints on the strategies are in terms of the distributions $\underline{p}_t$ used. This leads to the specific constrained minimum on $\liminf R_T / \ln T$ stated in Lemma 4.

## B.1 General lower bound given by a constrained infimum

For a model $\mathcal{D}$ and a bandit problem $\underline{\nu}$ in $\mathcal{D}$ with a single optimal distribution $\underline{p}^\star(\underline{\nu})$ in $\mathcal{P}$, we introduce the following set of confusing alternative bandit problems:

$$\mathrm{ALT}(\underline{\nu}) = \left\{ \underline{\nu}' \text{ in } \mathcal{D} \ \middle| \ p^\star(\underline{\nu}) \notin \mathrm{Opt}(\underline{\nu}') \ \text{ and } \ \forall\, 1 \leqslant a \leqslant K, \qquad \nu_a = \nu_a' \right.$$
$$\left. \text{or} \quad \left[ p_a^\star(\underline{\nu}) = 0 \ \text{ and } \ \mathrm{KL}(\nu_a, \nu_a') < +\infty \right] \right\}.$$

The bandit problems in $\mathrm{ALT}(\underline{\nu})$ are such that $p^\star(\underline{\nu})$ is suboptimal for them, yet, the player cannot discriminate them from $\underline{\nu}$ by only playing $p^\star(\underline{\nu})$, as, for each arm $a$, either $p_a^\star(\underline{\nu}) = 0$ and selecting the optimal probability $\underline{p}^\star(\underline{\nu})$ never results in picking arm $a$, or $\nu_a = \nu_a'$ and observing a reward associated with $a$ does not provide discriminative information.

The proof of the following lemma relies on standard techniques introduced by Graves & Lai (1997). This is why we postpone its proof to Appendix B.3. The linear program defining $c\big(\mathrm{Ext}(\mathcal{P}), \underline{\nu}\big)$ is over vectors $(n_{\underline{p}})$, where $\underline{p} \neq \underline{p}^\star(\underline{\nu})$, i.e., no component $n_{\underline{p}^\star(\underline{\nu})}$ is considered.

**Lemma 4.** *For all possibly randomized UFC strategy over $\mathcal{D}$ given $\mathcal{P}$, only picking distributions in $\mathrm{Ext}(\mathcal{P})$, for all bandit problems $\underline{\nu}$ in $\mathcal{D}$ with a single optimal distribution in $\mathcal{P}$ denoted by $\underline{p}^\star(\underline{\nu})$,*

$$\liminf_{T \to \infty} \frac{R_T}{\ln T} \geqslant c\big(\mathrm{Ext}(\mathcal{P}), \underline{\nu}\big)$$

$$\stackrel{\text{def}}{=} \inf \left\{ \sum_{\substack{\underline{p} \in \mathrm{Ext}(\mathcal{P}) \\ \underline{p} \neq \underline{p}^\star(\underline{\nu})}} \Delta_{\underline{p}}\, n_{\underline{p}} : \ n_{\underline{p}} \geqslant 0 \ \text{ and } \ \forall\, \underline{\nu}' \in \mathrm{ALT}(\underline{\nu}), \ \sum_{\substack{\underline{p} \in \mathrm{Ext}(\mathcal{P}) \\ \underline{p} \neq \underline{p}^\star(\underline{\nu})}} n_{\underline{p}} \sum_{\substack{a \in [K] \\ p_a^\star(\underline{\nu}) = 0}} p_a\, \mathrm{KL}(\nu_a, \nu_a') \geqslant 1 \right\}.$$

We may also prove the following equivalence, which will be the key for both the proof of Theorem 2 and the one of Proposition 1.

**Lemma 5.** *Consider a (mean-bounded or mean-unbounded) model $\mathcal{D}$, a diversity-preserving polytope $\mathcal{P}$ and a bandit problem $\underline{\nu}$ in $\mathcal{D}$ with a single optimal distribution $\underline{p}^\star(\underline{\nu})$ in $\mathcal{P}$. We have the equivalence:*

$$\mathrm{ALT}(\underline{\nu}) \neq \varnothing \qquad \Longleftrightarrow \qquad c\big(\mathrm{Ext}(\mathcal{P}), \underline{\nu}\big) > 0\,.$$

*Proof.* If $\mathrm{ALT}(\underline{\nu})$ is empty, then the linear program defining $c\big(\mathrm{Ext}(\mathcal{P}), \underline{\nu}\big)$ is unconstrained, so that $c\big(\mathrm{Ext}(\mathcal{P}), \underline{\nu}\big) = 0$. If $\mathrm{ALT}(\underline{\nu})$ contains a problem $\underline{\nu}'$, then since $p^\star(\underline{\nu})$ is not optimal for $\underline{\nu}'$, there exists at least one $a \in [K]$ such that $p_a^\star(\underline{\nu}) = 0$ and $\nu_a' \neq \nu_a$ with $\mathrm{KL}(\nu_a, \nu_a') < +\infty$, and we also have

$$\mathcal{K}_{\max}(\underline{\nu}, \underline{\nu}') \stackrel{\text{def}}{=} \max_{\substack{a \in [K] \\ p_a^\star(\underline{\nu}) = 0}} \mathrm{KL}(\nu_a, \nu_a') < +\infty\,.$$

Thus, by the constraint satisfied by $\underline{\nu}'$ for the first inequality and for the second equality, by substituting $1 \leqslant \Delta_{\underline{p}}/\Delta$ for $\underline{p} \neq \underline{p}^\star(\underline{\nu})$, we get

$$1 \leqslant \sum_{\substack{\underline{p} \in \mathrm{Ext}(\mathcal{P}) \\ \underline{p} \neq \underline{p}^\star(\underline{\nu})}} n_{\underline{p}} \sum_{\substack{a \in [K] \\ p_a^\star(\underline{\nu}) = 0}} \underbrace{p_a}_{\leqslant 1}\, \underbrace{\mathrm{KL}(\nu_a, \nu_a')}_{\leqslant \mathcal{K}_{\max}(\underline{\nu}, \underline{\nu}')} \leqslant \frac{\mathcal{K}_{\max}(\underline{\nu}, \underline{\nu}')}{\Delta} \sum_{\substack{\underline{p} \in \mathrm{Ext}(\mathcal{P}) \\ \underline{p} \neq \underline{p}^\star(\underline{\nu})}} \Delta_{\underline{p}}\, n_{\underline{p}}\,.$$

This imposes the lower bound $\Delta/\mathcal{K}_{\max}(\underline{\nu}, \underline{\nu}') > 0$ on $c\big(\mathrm{Ext}(\mathcal{P}), \underline{\nu}\big)$. $\qquad \square$

## B.2 Proof of Theorem 2

**Step 1: Reduction argument.** We first note that any (possibly randomized) strategy picking distributions in the polytope $\mathcal{P}$ can be converted into a (randomized) strategy picking distributions only in $\text{Ext}(\mathcal{P})$, as required by Lemma 4. This is indeed possible as only the final pure actions drawn matter. More precisely, given a probability distribution $\rho_t$ over $\mathcal{P}$, we introduce

$$\int_{\mathcal{P}} \underline{p} \, d\rho_t(\underline{p}) = \sum_{\underline{p} \in \text{Ext}(\mathcal{P})} \Phi_{\underline{p}}(\rho_t) \, \underline{p}, \qquad \text{and let} \quad \Phi(\rho_t) = \left(\Phi_{\underline{p}}(\rho_t)\right),$$

where the convex weights $\Phi_{\underline{p}}(\rho_t)$ exist as $\mathcal{P}$ is the convex hull of $\text{Ext}(\mathcal{P})$. We interpret the vector $\Phi(\rho_t)$ as a probability distribution over $\text{Ext}(\mathcal{P})$. Now, the random variables $A_t \in [K]$ and $A_t' \in [K]$ drawn as follows, in two-stage randomizations, have the same distributions:

$$\underline{p}_t \sim \rho_t \text{ then } A_t \sim \underline{p}_t \qquad \text{and} \qquad \underline{p}_t' \sim \Phi(\rho_t) \text{ then } A_t' \sim \underline{p}_t'.$$

**Step 2: Application of the results of Section B.1.** Thus, by Lemmas 4 and 5, it suffices to show that $\text{ALT}(\underline{\nu})$ is not empty. Let $a \in [K]$ such that $p_a^\star(\underline{\nu}) = 0$. By assumption, $\mathcal{P}$ thus $\text{Ext}(\mathcal{P})$ put some probability mass on this arm $a$: there exists $\underline{p} \in \text{Ext}(\mathcal{P})$ with $p_a > 0$. Since $\underline{p}^\star(\nu)$ is the unique optimal arm of $\nu$, we have $\Delta_{\underline{p}} > 0$. Now, by the assumption of unbounded means in $\mathcal{D}$, there exists a distribution $\zeta_a \in \mathcal{D}$ with expectation $> \mu_a + \Delta(\underline{p})/p_a$. By Assumption 1, there actually exists $\nu_a' \in \mathcal{D}$ with expectation $\mu_a' > \mu_a + \Delta(\underline{p})/p_a$ and such that $\text{KL}(\nu_a, \nu_a') < +\infty$.

We denote by $\underline{\nu}'$ the bandit problem such that $\nu_k' = \nu_k$ for all $k \neq a$, and whose $a$–th distribution is $\nu_a'$, and claim that $\underline{\nu}' \in \text{ALT}(\underline{\nu})$. To see this, it only remains to show that $\underline{p}^\star(\underline{\nu})$ is suboptimal for $\underline{\nu}'$; indeed, since $p_a^\star(\underline{\nu}) = 0$ while $\underline{\nu}$ and $\underline{\nu}'$ only differ at $a$ for the first equality, by definition of $\Delta_{\underline{p}}$ for the second equality, and by construction $\nu_a'$ for the rest,

$$\langle \underline{p}^\star(\underline{\nu}), \, \underline{\mu}' \rangle = \langle \underline{p}^\star(\underline{\nu}), \, \underline{\mu} \rangle = \langle \underline{p}, \, \underline{\mu} \rangle + p_a \frac{\Delta_{\underline{p}}}{p_a} < \langle \underline{p}, \, \underline{\mu}' \rangle.$$

## B.3 Proof of Lemma 4

We believe that we offer a neater proof than what is usually proposed in the literature.

We fix a possibly randomized UFC strategy over $\mathcal{D}$ given $\mathcal{P}$, only picking distributions in $\text{Ext}(\mathcal{P})$, and a bandit problem $\underline{\nu}$ in $\mathcal{D}$ with a single optimal distribution $\underline{p}^\star(\underline{\nu})$. We know that the correct scaling of the suboptimal pulls is at most logarithmic and therefore define the normalized allocations, for all $\underline{p} \in \text{Ext}(\mathcal{P})$,

$$n_{T,\underline{p}} = \frac{\mathbb{E}_{\underline{\nu}}\left[N_{\underline{p}}(T)\right]}{\ln T}, \qquad \text{so that} \qquad \frac{R_T}{\ln T} = \sum_{\underline{p} \in \mathcal{P}} \Delta_{\underline{p}} \frac{\mathbb{E}_{\underline{\nu}}\left[N_{\underline{p}}(T)\right]}{\ln T} = \sum_{\underline{p} \in \mathcal{P}} \Delta_{\underline{p}} \, n_{T,\underline{p}}. \tag{20}$$

A UFC algorithm facing the problem $\underline{\nu}$ will eventually focus on the unique optimal distribution $\underline{p}^\star(\underline{\nu})$. Thus, most of its observations will correspond to pure actions $a \in [K]$ such that $p_a^\star(\underline{\nu}) > 0$, which provide no useful information to distinguish $\underline{\nu}$ from a given confusing alternative problem $\underline{\nu}' \in \text{ALT}(\underline{\nu})$. A measure of the useful information gained is the Kullback-Leibler divergence

$$\mathcal{I}_T \stackrel{\text{def}}{=} \text{KL}\left(\mathbb{P}_{\underline{\nu},T}, \mathbb{P}_{\underline{\nu}',T}\right),$$

where $\mathbb{P}_{\underline{\nu},T}$ and $\mathbb{P}_{\underline{\nu}',T}$ denote the distributions of rewards $Y_1, \ldots, Y_T$ obtained in the first $T$ rounds (and of the auxiliary randomizations used) when the underlying problems are $\underline{\nu}$ and $\underline{\nu}'$, respectively.

**Step 1: Rewriting $\mathcal{I}_T$.** By a chain rule—see Equation (8) in Garivier et al. (2019)—for the first equality below, by an application of the tower rule for the second equality, and by grouping distributions output by

the strategy by their values in $\mathrm{Ext}(\mathcal{P})$,

$$\mathcal{I}_T = \sum_{t=1}^{T} \mathbb{E}_{\underline{\nu}}\big[\mathrm{KL}(\nu_{A_t}, \nu'_{A_t})\big] = \sum_{t=1}^{T} \mathbb{E}_{\underline{\nu}}\left[\sum_{a=1}^{K} p_{t,a}\,\mathrm{KL}(\nu_a, \nu'_a)\right] = \sum_{\underline{p} \in \mathrm{Ext}(\mathcal{P})} \mathbb{E}_{\underline{\nu}}\big[N_{\underline{p}}(T)\big] \sum_{a \in [K]} p_a\,\mathrm{KL}(\nu_a, \nu'_a). \quad (21)$$

Since alternative problems $\underline{\nu}' \in \mathrm{ALT}(\underline{\nu})$ are such that $\nu'_a = \nu_a$ when $p^{\star}_a(\underline{\nu}) > 0$, we finally obtain

$$\frac{\mathcal{I}_T}{\ln T} = \sum_{\substack{\underline{p} \in \mathrm{Ext}(\mathcal{P})}} n_{T,\underline{p}} \sum_{\substack{a \in [K] \\ p^{\star}_a(\underline{\nu})=0}} p_a\,\mathrm{KL}(\nu_a, \nu'_a) = \sum_{\substack{\underline{p} \in \mathrm{Ext}(\mathcal{P}) \\ \underline{p} \neq \underline{p}^{\star}(\underline{\nu})}} n_{T,\underline{p}} \sum_{\substack{a \in [K] \\ p^{\star}_a(\underline{\nu})=0}} p_a\,\mathrm{KL}(\nu_a, \nu'_a). \quad (22)$$

**Step 2: UFC entails that $\mathcal{I}_T$ is larger than $\ln T$ in the limit.** This step is a mere adaptation of the standard proof technique by Lai & Robbins (1985) and Garivier et al., 2019, with distributions in $\mathrm{Ext}(\mathcal{P})$ playing the role of arms in the mentioned references. More formally, the crucial observation is that the regret is expressed in (2) in terms of the number of plays of suboptimal distributions in $\mathrm{Ext}(\mathcal{P})$. Thus, that the strategy is UFC and that $\underline{p}^{\star}(\underline{\nu})$ is, by definition of $\mathrm{ALT}(\underline{\nu})$, suboptimal for $\underline{\nu}'$ entail that, for all $\alpha > 0$,

$$\forall \underline{p} \neq \underline{p}^{\star}(\underline{\nu}), \quad \mathbb{E}_{\underline{\nu}}\big[N_{\underline{p}}(T)\big] = o(T^{\alpha}), \qquad \text{as well as} \qquad \mathbb{E}_{\underline{\nu}'}\big[N_{\underline{p}^{\star}(\underline{\nu})}(T)\big] = o(T^{\alpha}). \quad (23)$$

In particular, for all $\varepsilon > 0$, there exists $T$ large enough so that $\mathbb{E}_{\underline{\nu}'}\big[N_{\underline{p}^{\star}(\underline{\nu})}(T)\big] \leqslant T^{\varepsilon}$. We denote by $\mathrm{kl}(p,q) = p\ln(p/q) + (1-p)\ln\big((1-p)/(1-q)\big)$ the Kullback-Leibler divergence between two Bernoulli distributions with parameters $p$ and $q$. By the data-processing inequality for $[0,1]$–valued random variables (see Section 2.1 in Garivier et al., 2019) and by the standard inequality $\mathrm{kl}(p,q) \geqslant p\ln(1/q) - \ln 2$, for all $\varepsilon > 0$, for all $T$ sufficiently large,

$$\mathcal{I}_T = \mathrm{KL}\big(\mathbb{P}_{\underline{\nu},T}, \mathbb{P}_{\underline{\nu}',T}\big) \geqslant \mathrm{kl}\left(\mathbb{E}_{\underline{\nu}}\left[\frac{N_{\underline{p}^{\star}(\underline{\nu})}(T)}{T}\right], \mathbb{E}_{\underline{\nu}'}\left[\frac{N_{\underline{p}^{\star}(\underline{\nu})}(T)}{T}\right]\right) \geqslant \underbrace{\mathbb{E}_{\underline{\nu}}\left[\frac{N_{\underline{p}^{\star}(\underline{\nu})}(T)}{T}\right]}_{\longrightarrow 1} \ln\left(\underbrace{\frac{T}{\mathbb{E}_{\underline{\nu}'}\big[N_{\underline{p}^{\star}(\underline{\nu})}(T)\big]}}_{\geqslant T^{1-\varepsilon}}\right) - \ln 2\,,$$

where we substituted (23). Letting $T \to \infty$ then $\varepsilon \to 0$, we conclude to

$$\liminf_{T \to \infty} \frac{\mathcal{I}_T}{\ln T} \geqslant \sup_{\varepsilon > 0} \liminf_{T \to \infty} \frac{\ln T^{1-\varepsilon}}{\ln T} = 1\,. \quad (24)$$

**Remark 1.** *The specificity of the setting considered appears in these first two steps: we use the UFC property on $\mathrm{Ext}(\mathcal{P})$ but pure actions $A_t$ are eventually played, so that we could lower bound $\mathcal{I}_T/\ln T$ in (22) by*

$$\sum_{\substack{\underline{p} \in \mathrm{Ext}(\mathcal{P})}} n_{T,\underline{p}} \sum_{\substack{a \in [K] \\ p^{\star}_a(\underline{\nu})=0}} p_a\,\mathrm{KL}(\nu_a, \nu'_a) \geqslant \sum_{\substack{\underline{p} \in \mathrm{Ext}(\mathcal{P})}} n_{T,\underline{p}}\,\mathrm{KL}\left(\sum_{a \in [K]} p_a\,\nu_a, \sum_{a \in [K]} p_a\,\nu'_a\right),$$

*where the inequality holds by convexity of $\mathrm{KL}$ and the fact that $\nu'_a = \nu_a$ when $p^{\star}_a(\underline{\nu}) > 0$. The right-hand side above corresponds to the measure of information gained when playing distributions $\underline{p}_t$ without observing the pure actions $A_t$.*

**Step 3: Considering cluster points.** We combine (20), (22), and (24) as follows. Let $c$ be a cluster point of the sequence $R_T/\ln T$. If $c = +\infty$ is the only value, then $R_T/\ln T \to +\infty$ and the result is proved; otherwise, take a finite $c$. We denote by $(T_m)_{m \geqslant 1}$ an increasing sequence such that $R_{T_m}/\ln T_m \to c$. In view of the decomposition (20) and since $n_{T_m,\underline{p}} \geqslant 0$ and $\Delta(\underline{p}) > 0$ for all $\underline{p} \in \mathrm{Ext}(\mathcal{P})$ with $\underline{p} \neq \underline{p}^{\star}(\underline{\nu})$, all these sequences $n_{T_m,\underline{p}}$ are bounded. Hence, we may extract a subsequence $(T_{m_k})_{k \geqslant 1}$ from $(T_m)$ such that all sequences $n_{T_{m_k},\underline{p}}$ converge as $k \to \infty$, to limits denoted by $n_{\underline{p}}$. This only holds for $\underline{p} \in \mathrm{Ext}(\mathcal{P})$ with $\underline{p} \neq \underline{p}^{\star}(\underline{\nu})$, but $c\big(\mathrm{Ext}(\mathcal{P}), \underline{\nu}\big)$ is only defined based on such vectors.

These convergences yield first, by (20), that

$$\liminf_{T\to\infty} \frac{R_T}{\ln T} \geqslant \sum_{\substack{\underline{p}\in\mathrm{Ext}(\mathcal{P}) \\ \underline{p}\neq\underline{p}^\star(\underline{\nu})}} \Delta_{\underline{p}}\, n_{\underline{p}}\,.$$

These convergences also entail, together with (22) and (24), and the definition of $\liminf$, that, for any $\underline{\nu}' \in \mathrm{ALT}(\underline{\nu})$,

$$\sum_{\substack{\underline{p}\in\mathrm{Ext}(\mathcal{P}) \\ \underline{p}\neq\underline{p}^\star(\underline{\nu})}} n_{\underline{p}} \sum_{\substack{a\in[K] \\ p_a^\star(\underline{\nu})=0}} p_a\,\mathrm{KL}(\nu_a,\nu_a') \geqslant \liminf_{T\to+\infty} \sum_{\substack{\underline{p}\in\mathrm{Ext}(\mathcal{P}) \\ \underline{p}\neq\underline{p}^\star(\underline{\nu})}} n_{T,\underline{p}} \sum_{\substack{a\in[K] \\ p_a^\star(\underline{\nu})=0}} p_a\,\mathrm{KL}(\nu_a,\nu_a') \geqslant 1\,.$$

Put differently, the limits $n_{\underline{p}}$ defined above, where $\underline{p}\neq\underline{p}^\star(\underline{\nu})$, satisfy the constraints in the defining infimum of $c\big(\mathrm{Ext}(\mathcal{P}),\underline{\nu}\big)$. This concludes the proof.

## C   Proof of Proposition 1

The proof builds on the proofs for upper bounds (Appendix A) and lower bounds (Appendix B).

We provide here some general considerations. The Bernoulli distributions are sub-Gaussian with parameter $\sigma^2 = 1/4$. The problem $\underline{\nu} = \big(\mathrm{Ber}(0),\mathrm{Ber}(1/2),\mathrm{Ber}(0)\big)$ considered has expectations $\underline{\mu} = (0,1/2,0)$. The distributions $\underline{p}^{(1)} = (0,1/2,1/2)$ and $\underline{p}_\delta^{(2)} = (\delta,0,1-\delta)$ obtain respective expected rewards

$$\big\langle \underline{p}^{(1)},\,\underline{\mu} \big\rangle = \frac{1}{4} > 0 = \big\langle \underline{p}_\delta^{(2)},\,\underline{\mu} \big\rangle\,.$$

In particular, given that the polytope $\mathcal{P}_\delta$ considered is the segment between $\underline{p}^{(1)}$ and $\underline{p}_\delta^{(2)}$, the distribution $\underline{p}^{(1)}$ is the unique optimal distribution in $\mathcal{P}$.

### C.1   Case $\delta > 1/4$: a $\ln T$ regret is suffered

The setting of Proposition 1 satisfies the assumptions of Lemmas 4 and 5, up to considering the reduction performed in Appendix B.2. It therefore suffices to show that $\mathrm{ALT}(\underline{\nu})$ is not empty.

Given that $\underline{p}^{(1)}$ puts a positive probability mass on arms 2 and 3, alternative problems in $\mathrm{ALT}(\underline{\nu})$ may differ from $\underline{\nu}$ only at arm 1. Given that $\mathrm{KL}\big(\mathrm{Ber}(0),\mathrm{Ber}(\mu_1)\big) < +\infty$ if and only if $\mu_1 \in [0,1)$, we have

$$\mathrm{ALT}(\underline{\nu}) = \Big\{ \underline{\nu}' = \big(\mathrm{Ber}(\mu_1'),\mathrm{Ber}(1/2),\mathrm{Ber}(0)\big): \quad \mu_1' \in [0,1) \text{ and } \underline{p}^{(1)} \notin \mathrm{Opt}(\underline{\nu}') \Big\}\,.$$

Now, denoting by $\underline{\mu}'$ the expectations of $\underline{\nu}'$, and given that $\mathcal{P}_\delta$ is a segment, we have that the condition $\underline{p}^{(1)} \notin \mathrm{Opt}(\underline{\nu}')$ is equivalent to

$$\big\langle \underline{p}^{(1)},\,\underline{\mu}' \big\rangle = 1/4 < \delta\,\mu_1' = \big\langle \underline{p}^{(2)},\,\underline{\mu}' \big\rangle\,,$$

which is equivalent to $\mu_1' > 1/(4\delta)$. Therefore,

$$\mathrm{ALT}(\underline{\nu}) = \Big\{ \underline{\nu}' = \big(\mathrm{Ber}(\mu_1'),\mathrm{Ber}(1/2),\mathrm{Ber}(0)\big): \quad 1/(4\delta) < \mu_1' < 1 \Big\}\,.$$

$\mathrm{ALT}(\underline{\nu})$ is not empty if and only if $\delta > 1/4$, which concludes the proof.

### C.2   Case $\delta < 1/4$: a bounded regret may be suffered

We use the Box-B algorithm with $u_0 = 1/2$ but make sure that the upper-confidence estimates of the expectations are always smaller than the known bound 1 on rewards, i.e., we replace Step 1 of Box B by

$$V_a(t-1) = \min\big\{ U_a(t-1),\,1 \big\}\,, \qquad \underline{V}(t-1) = \big(V_1(t),\dots,V_K(t)\big)\,, \qquad \underline{p}_t \in \underset{\underline{p}\in\mathrm{Ext}(\mathcal{P})}{\mathrm{argmax}}\, \big\langle \underline{p},\,\underline{V}(t-1) \big\rangle\,;$$

this is the crucial modification for this proof, see (26) for an explanation why it is handy. All other steps of the Box-B algorithm remain unchanged.

**Step 1: Preparation and adaptation of existing proofs.** We borrow several elements from the proof conducted in Appendix A. We consider the same favorable event $\mathcal{E}(t)$ as therein, but only one event in terms of number of pulls—namely, we are only interested in arm 3:

$$\mathcal{E}(t) = \left\{ \text{For all } a \in [K], \quad |\mu_a - \widehat{\mu}_a(t)| \leqslant \sqrt{\frac{2\ln t}{\max\{N_a(t), 1\}}} \right\} \qquad \text{and} \qquad \mathcal{G}(t) = \left\{ N_3(t) \geqslant t/3 \right\}.$$

Lemma 3 shows that $\mathbb{P}\big(\overline{\mathcal{E}(t)}\big) \leqslant 2Kt^{-3}$ for all $t \geqslant 2 > e^{1/8}$. Given that $p_{t,3} \geqslant \min\{p_3^{(1)}, p_{\delta,3}^{(2)}\} = 1/2$ by definition of $\mathcal{P}_\delta$ as a segment and the fact that $\delta < 1/4$, we apply the Hoeffding–Azuma bound (18) as follows: for all $t \geqslant 1$,

$$\mathbb{P}\big(\overline{\mathcal{G}(t)}\big) \leqslant \mathbb{P}\left\{ N_3(t) - \frac{t}{2} \leqslant -\frac{t}{6} \right\} \leqslant \mathbb{P}\left\{ N_3(t) - \sum_{s=1}^t p_{s,3} \leqslant -\frac{t}{6} \right\} \leqslant e^{-t/18}.$$

Since $\delta < 1/4$, there exists a threshold $t_0 \geqslant 2$ such that

$$\forall t \geqslant t_0, \qquad \delta + 2\sqrt{\frac{6\ln t}{t}} < \frac{1}{4}.$$

We show below that

$$\forall t \geqslant t_0, \quad \text{on } \mathcal{E}(t) \cap \mathcal{G}(t), \qquad \big\langle \underline{p}^{(1)}, \underline{V}(t) \big\rangle > \big\langle \underline{p}_\delta^{(2)}, \underline{V}(t) \big\rangle, \qquad \text{thus,} \qquad \underline{p}_{t+1} = \underline{p}^{(1)}. \qquad (25)$$

We recall that $\underline{p}^{(1)}$ is the unique optimal distribution for $\underline{\nu}$ and that the gap of $\underline{p}_\delta^{(2)}$ equals $1/4$. The property (25) therefore leads to the following bounded regret bound: for all $T \geqslant t_0$,

$$R_T = \frac{1}{4} \sum_{t=1}^T \mathbb{P}\{\underline{p}_t = \underline{p}_\delta^{(2)}\} \leqslant \frac{t_0}{4} + \frac{1}{4} \sum_{t=t_0}^{+\infty} \mathbb{P}\big(\overline{\mathcal{E}(t)} \cup \overline{\mathcal{G}(t)}\big) \leqslant \frac{t_0}{4} + \frac{1}{4} \sum_{t=t_0}^{+\infty} \big(2Kt^{-3} + e^{-t/18}\big) = \frac{t_0}{4} + \frac{K}{2} + 5.$$

**Step 2: Proving** (25). By definition, we have, $\big\langle \underline{p}^{(1)}, \underline{V}(t) \big\rangle \geqslant \big\langle \underline{p}^{(1)}, \underline{\nu} \big\rangle = 1/4$ on $\mathcal{E}(t)$, and

$$\text{still on } \mathcal{E}(t), \qquad \big\langle \underline{p}_\delta^{(2)}, \underline{V}(t) \big\rangle = \delta \underbrace{V_1(t)}_{\leqslant 1} + (1-\delta) \underbrace{V_3(t)}_{\leqslant 1} \leqslant \delta + 2\sqrt{\frac{2\ln t}{\max\{N_3(t), 1\}}}. \qquad (26)$$

We crucially used here the boundedness of $V_1(t)$, which we upper bounded by 1. This is fortunate, as arm 1 is only played when the suboptimal distribution $\underline{p}_\delta^{(2)}$ is picked, so that $N_1(t)$ could be small and $\mathcal{E}(t)$ would not provide an efficient bound.

Therefore,

$$\text{on } \mathcal{E}(t) \cap \mathcal{G}(t), \qquad \big\langle \underline{p}^{(1)}, \underline{V}(t) \big\rangle \geqslant \frac{1}{4} \qquad \text{and} \qquad \big\langle \underline{p}_\delta^{(2)}, \underline{V}(t) \big\rangle \leqslant \delta + 2\sqrt{\frac{6\ln t}{t}},$$

so that (25) follows by the definition of $t_0$.

## D   Proof of Theorem 3

The proof strategy is similar to that of Theorem 1 with some differences we will highlight: we still work under the favorable event under which all the confidence bounds are correct, and all arms have been picked a linear number of times. Both these events hold with high probability since the probability of picking any arm is lower bounded by some positive constant at all rounds. In that case, the upper confidence vectors $\underline{U}(t)$ are a good estimators of the true mean payoff vector $\underline{\mu}$, and the distributions picked by diversity-preserving UCB get close to the optimal distribution(s). Crucially, we show in Lemma 7 that the differences between

the expected payoffs of the distributions selected by UCB and the optimal payoff depend *quadratically* on the differences between $\underline{U}(t)$ and $\underline{\mu}$. This is a consequence of the curvature of the diversity-preserving set $\mathcal{P}_r$. Since the error in estimating $\underline{\mu}$ by $\underline{U}(t)$ is of order $\sqrt{\ln t / t}$, we obtain per-round regrets of order $\ln t / t$, which sum up to $\ln^2 T$.

We start with two lemmas, providing first a closed-form expression of the optimal distribution (Lemma 6) and second, deducing a quadratic upper bound on the instantaneous regret (Lemma 7).

Denote by $V = \left\{ \underline{\mu} \in \mathbb{R}^d \mid \mu_1 = \cdots = \mu_K \right\}$ the subset of expected-payoff vectors with identical components. All strategies get a null regret in bandit problems $\underline{\nu}$ with payoffs in $V$. We may therefore exclude this case in the rest of the proof. We denote by $\mathbf{1}$ the vector in $\mathbb{R}^K$ with components all equal to 1, and refer to the model of all distributions over $\mathbb{R}$ with a finite first moment as $\mathcal{D}_{\text{all}}$.

**Lemma 6.** *Consider $\mathcal{P}_r$ with $r < r_{\lim}$. For any bandit problem $\underline{\nu}$ in $\mathcal{D}_{\text{all}}$ with means $\underline{\mu} \in \mathbb{R}^K \setminus V$, the optimal distribution in $\mathcal{P}_r$ is unique, only depends on $\underline{\mu}$, and equals*

$$\underline{p}_\star(\underline{\mu}) = \frac{1}{K}\mathbf{1} + \frac{r}{\|\underline{\mu} - \mu_{\text{avg}}\mathbf{1}\|}\left(\underline{\mu} - \mu_{\text{avg}}\mathbf{1}\right), \qquad where \qquad \mu_{\text{avg}} = \frac{1}{K}\sum_{a \in [K]} \mu_a.$$

*Proof.* We denote already by $\underline{p}_\star(\underline{\mu})$ the vector defined above and prove that it is the only optimal vector; we do so via elementary arguments. For any $\underline{p}$ in $\mathcal{P}_r$, the difference between the expected payoffs of $\underline{p}$ and $\underline{p}_\star(\underline{\mu})$ equals

$$
\begin{aligned}
\left\langle \underline{\mu}, \underline{p}_\star(\underline{\mu}) - \underline{p} \right\rangle &= \left\langle \underline{\mu}, \frac{1}{K}\mathbf{1} - \underline{p} \right\rangle + \frac{r}{\|\underline{\mu} - \mu_{\text{avg}}\mathbf{1}\|}\left\langle \underline{\mu}, \underline{\mu} - \mu_{\text{avg}}\mathbf{1} \right\rangle \\
&= \left\langle \underline{\mu} - \mu_{\text{avg}}\mathbf{1}, \frac{1}{K}\mathbf{1} - \underline{p} \right\rangle + \frac{r}{\|\underline{\mu} - \mu_{\text{avg}}\mathbf{1}\|}\left\langle \underline{\mu} - \mu_{\text{avg}}\mathbf{1}, \underline{\mu} - \mu_{\text{avg}}\mathbf{1} \right\rangle \\
&\geqslant \|\underline{\mu} - \mu_{\text{avg}}\mathbf{1}\|\left(r - \left\|\frac{1}{K}\mathbf{1} - \underline{p}\right\|\right) \geqslant 0,
\end{aligned}
\tag{27}
$$

where we used, for the second equality, that both vectors $(1/K)\mathbf{1} - \underline{p}$ and $\underline{\mu} - \mu_{\text{avg}}\mathbf{1}$ are orthogonal to $\mathbf{1}$ (as their components sum up to 0), and where, for the final inequality, we applied the the Cauchy-Schwarz inequality:

$$\left\langle \underline{\mu} - \mu_{\text{avg}}\mathbf{1}, \frac{1}{K}\mathbf{1} - \underline{p} \right\rangle \geqslant -\|\underline{\mu} - \mu_{\text{avg}}\mathbf{1}\| \cdot \left\|\frac{1}{K}\mathbf{1} - \underline{p}\right\|.$$

Now, the inequality (27) is an equality if and only if $\underline{\mu} - \mu_{\text{avg}}\mathbf{1}$ and $(1/K)\mathbf{1} - \underline{p}$ are colinear, that is, if there exists $\alpha \in \mathbb{R}$ such that

$$\underline{p} = \frac{1}{K}\mathbf{1} + \frac{\alpha}{\|\underline{\mu} - \mu_{\text{avg}}\mathbf{1}\|}\left(\underline{\mu} - \mu_{\text{avg}}\mathbf{1}\right).$$

By definition of $\mathcal{P}_r$, such a $\underline{p}$ is in $\mathcal{P}_r$ if and only if $|\alpha| \leqslant r$. Therefore, a final equality to 0 is achieved in (27) if and only if $\alpha = r$, i.e., if $\underline{p} = \underline{p}_\star(\underline{\mu})$. $\square$

The curvature of the diversity-preserving set $\mathcal{P}_r$ guarantees that the best distribution varies smoothly with the expected payoffs. This in turn implies that the per-round regret of the diversity-preserving UCB strategy depends quadratically on the difference between the upper confidence bound vector $\underline{U}$ and $\underline{\mu}$—at least when $\underline{U}$ is in a neighborhood of $\underline{\mu}$.

**Lemma 7.** *Consider $\mathcal{P}_r$ with $r < r_{\lim}$ and use the notation of Lemma 6. For any expected-payoff vector $\underline{\mu} \in \mathbb{R}^d \setminus V$, define the function*

$$\phi_{\underline{\mu}} : \underline{v} \in \mathbb{R}^K \setminus V \longmapsto \left\langle \underline{\mu}, \underline{p}_\star(\underline{v}) \right\rangle.$$

*There exist a constant $B_{\underline{\mu}}$ such that*

$$\forall \underline{v} \in \mathbb{R}^K \setminus V, \qquad \phi_{\underline{\mu}}(\underline{\mu}) - \phi_{\underline{\mu}}(\underline{v}) \leqslant B_{\underline{\mu}}\|\underline{\mu} - \underline{v}\|^2.$$

With this notation, if $\underline{p}_t$ is the distribution output by the diversity-preserving UCB strategy at round $t$, then the expected per-round regret $r_t$ suffered at time $t$ can be expressed as

$$r_t = \mathbb{E}\Big[\big\langle \underline{\mu},\, \underline{p}_\star(\underline{\mu}) - \underline{p}_t \big\rangle\Big] = \phi_{\underline{\mu}}(\underline{\mu}) - \mathbb{E}\Big[\phi_{\underline{\mu}}\big(\underline{U}(t-1)\big)\Big].$$

*Proof.* By Lemma 6, $\phi_{\underline{\mu}}$ admits the closed-form expression

$$\phi_{\underline{\mu}} : \underline{v} \in \mathbb{R}^K \setminus V \longmapsto \frac{1}{K}\langle \underline{\mu}, \mathbf{1}\rangle + \frac{r}{\|\underline{v} - \underline{v}_{\mathrm{avg}}\mathbf{1}\|}\langle \underline{\mu},\, \underline{v} - \underline{v}_{\mathrm{avg}}\mathbf{1}\rangle,$$

showing that $\phi_{\underline{\mu}}$ is $\mathcal{C}^3$ at all points $\underline{v}$ such that $\underline{v} - v_{\mathrm{avg}}\mathbf{1} \neq 0$, i.e., over $\mathbb{R}^K \setminus V$. Moreover, by definition of $\underline{p}_\star(\underline{\mu})$, the function $\phi_{\underline{\mu}}$ reaches its maximum at $\underline{\mu}$, therefore its differential at $\underline{\mu}$ is null and the Taylor expansion around $\underline{\mu}$ of $\phi_{\underline{\mu}}$ reads

$$\phi_{\underline{\mu}}(\underline{v}) = \phi_{\underline{\mu}}(\underline{\mu}) + \mathcal{O}\big(\|\underline{v} - \underline{\mu}\|^2\big).$$

Put differently, the function

$$\psi_{\underline{\mu}} : \underline{v} \in \mathbb{R}^K \setminus V \longmapsto \frac{\phi_{\underline{\mu}}(\underline{v}) - \phi_{\underline{\mu}}(\underline{\mu})}{\|\underline{v} - \underline{\mu}\|^2}$$

is bounded in some $\varepsilon$–neighbourhood of $\underline{\mu}$. Outside of this neighborhood, the denominator $\|\underline{v} - \underline{\mu}\|$ is larger than $\varepsilon$ while the numerator is bounded by $2\max_{a \in [K]} \mu_a$. Therefore, $\psi_{\underline{\mu}}$ is bounded over the entire $\mathbb{R}^K \setminus V$. $\qquad\square$

We are now equipped to prove Theorem 3. The fact that the model $\mathcal{D}$ is composed of $\sigma^2$–sub-Gaussian distributions is used now. We adapt several arguments already reviewed in Appendix A and therefore provide a concise proof.

*Proof.* Consider the favorable events

$$\mathcal{E}(t) = \left\{\forall a \in [K], \quad |\mu_a - \widehat{\mu}_a(t)| \leqslant \sqrt{\frac{8\sigma^2 \ln t}{\max\{N_a(t), 1\}}}\right\} \quad \text{and} \quad \mathcal{G}'(t) = \left\{\forall a \in [K], \quad N_a(t) \geqslant \frac{t(r_{\mathrm{lim}} - r)}{2}\right\}.$$

For any $t$, under $\mathcal{E}(t) \cap \mathcal{G}'(t)$, by Lemma 7,

$$\big\langle \underline{\mu},\, \underline{p}_\star(\underline{\mu}) - \underline{p}_t \big\rangle = \phi_{\underline{\mu}}(\underline{\mu}) - \phi_{\underline{\mu}}\big(\underline{U}(t-1)\big) \leqslant B_{\underline{\mu}}\|\underline{\mu} - \underline{U}(t-1)\|^2 \leqslant B_{\underline{\mu}} \sum_{a \in [K]} \frac{16\sigma^2 \ln t}{t(r_{\mathrm{lim}} - r)} = \frac{16 K B_{\underline{\mu}}\sigma^2}{r_{\mathrm{lim}} - r}\frac{\ln t}{t}.$$

Therefore, taking expectations above,

$$r_t = \mathbb{E}\big[\big\langle \underline{\mu},\, \underline{p}_\star(\underline{\mu}) - \underline{p}_t \big\rangle\big] \leqslant \frac{16 K B_{\underline{\mu}}\sigma^2}{r_{\mathrm{lim}} - r}\frac{\ln t}{t} + \max_{a \in [K]} \mu_a\left(\mathbb{P}\big(\overline{\mathcal{E}(t)}\big) + \mathbb{P}\big(\overline{\mathcal{G}'(t)}\big)\right). \tag{28}$$

We showed already in (11) in the proof of Theorem 1 that $\displaystyle\sum_{t=1}^{+\infty} \mathbb{P}\big(\overline{\mathcal{E}(t)}\big) < +\infty$.

Similarly, as in the argument following (17), introduce the martingales $M_{a,t}$, and note that by (5), we have

$$\sum_{t=1}^{T} \mathbb{P}\big[\overline{\mathcal{G}'(t)}\big] \leqslant \sum_{t=1}^{T}\sum_{a \in [K]} \mathbb{P}\left\{M_{a,t} \leqslant -\frac{t(r_{\mathrm{lim}} - r)}{2}\right\}.$$

By the Azuma-Hoeffding inequality, the sum above is bounded, cf. (18). Sum (28) over $t$ to conclude. $\qquad\square$

