# OpenReview forum: "Diversity-Preserving $K$--Armed Bandits, Revisited"
_TMLR — Accepted by TMLR_

### Review · Reviewer_2E9z · 2024-04-20

**Summary Of Contributions:**

This paper gives upper and lower bounds for classes of multi-armed bandit problems with constraints on the action distributions. They discuss how such constraints can be used to describe various conditions that could be desirable in applications.

**Audience:**

Yes

**Claims And Evidence:**

No

**Requested Changes:**

The main thing to do is fix the technical errors.

A secondary, but very important change, is to do a more thorough literature review to see what has been done more recently related to these problems.

**Strengths And Weaknesses:**

# Strengths

* The problems posed are interesting and well-motivated.
* The writing is clear and well-structured.
* The results (if they can be shown to be true) are quite interesting, especially the bounded regret.

# Weaknesses

## Technical Correctness

The biggest weakness is that there is a mistake very early in the proof of Theorem 1. Namely, the inclusion of events described in (6) is incorrect. While many of the other steps in the analysis are correct, most results depend on (6) at some point, and so it is unclear which results in the paper are true.

We can see the incorrectness of (6) by a simple example. Say $K=2$ and $\mathcal{P}$ is just the standard probability simplex. (Any polytope which contains $\begin{bmatrix}0 \\\\1 \end{bmatrix}$ will suffice.) For any $\Delta >0$, $\sigma>0$, we can find $T$, $t$, and $n$ such that
* $1\le t \le T$
* $n\ge \frac{65 K\sigma^2}{\Delta}\ln T$
* $t\ge n/2$
* $\sqrt{8\sigma^2 \ln t}>\frac{\Delta}{2}$

For example, if $\Delta=\sigma=1$, $T=1000$, $n=899$, and $t=450$ satisfy all the constraints.

Say that $\underline{p}=\begin{bmatrix}0 \\\\ 1\end{bmatrix}$,  $N_1(t)=0$ and $N_2(t)=t$. Then the event on the left of (6) holds:
$$
\bigcap_{a \in [K]}\\{N_a(t)\ge np_a/2\\}
$$
The event on the right of (6) does not, since the right is
$$
\mathcal{E}'(t)=\left\\{\forall a\in [k], \sqrt{\frac{8\sigma^2 \ln t}{\max\\{N_a(t),1\\}}} < \frac{\Delta}{2}\right\\},
$$
and the inequality fails for $a=1$.

## Technical Precision

Beyond the error above, there are other instances of technical imprecision. For example, as written, **Assumption 1**  holds trivially, since we could always just take $\zeta=\zeta'$. Presumably something else was meant.

## Potentially Outdated Literature Review

While the problem is set up nicely, and context is given, the references are generally fairly old. From what I can gather from the text, this manuscript is based on a preprint written no later than 2022 (since it states that Liu et al. (2022) came after the preprint), and the bulk of the references come from prior to this time. My guess would be that many relevant follow-up works have already appeared. While the cited references may be the most relevant, giving an up-to-date discussion would be helpful.

I purposely did not look to see if the discussed works had more recent follow-up papers in order to preserve anonymity.

---

### Review · Reviewer_wpTj · 2024-04-21

**Summary Of Contributions:**

The paper studies the diversity-preserving K-armed bandits, a setting derived from the standard K-armed bandits in which the agent is allowed to play a distribution over the arms chosen in a properly provided set. The authors propose an optimistic regret minimization strategy that, under the assumption that the distribution set is a polytope, plays the extremal points distributions and uses the collected rewards to update the arm statistics. Under the subgaussian distributional assumption, the authors show that the algorithm enjoys logarithmic regret that reduces to constant under the assumption that the optimal distribution plays with non-zero probability each of the arms. A lower bound is provided matching the logarithmic regime. Furthermore, an example of the case in which the set of distributions is not a polytope is provided with a proposal on how to address it, leading to polylog regret.

**Audience:**

Yes

**Broader Impact Concerns:**

None.

**Claims And Evidence:**

Yes

**Requested Changes:**

Please refer to Weaknesses.

**Strengths And Weaknesses:**

**Strengths**
- The paper provides interesting results that are sound and quite expected, although I have not checked the proofs in detail.
- The setting considered, although introduced in previous works, seems to have nice applications in the real world.

**Weaknesses**
- [Related Works] The presented setting has notable connections with regret minimization with expert advice and bandits with mediator feedback. These settings share with the presented one the fact that arms are not directly pulled but a distribution (possibly non-stationary) is placed in the middle. In particular, these works are related since in some cases they are able to show that constant regret is possible, as in the present paper. The authors should include them in the related works section and provide a comparative discussion.

[1] Auer, Peter, Nicolo Cesa-Bianchi, Yoav Freund, and Robert E. Schapire. "The nonstochastic multiarmed bandit problem." SIAM journal on computing 32, no. 1 (2002): 48-77.

[2] Metelli, Alberto Maria, Matteo Papini, Pierluca D'Oro, and Marcello Restelli. "Policy optimization as online learning with mediator feedback." In Proceedings of the AAAI Conference on Artificial Intelligence, vol. 35, no. 10, pp. 8958-8966. 2021.

[3] Eldowa, Khaled, Nicolò Cesa-Bianchi, Alberto Maria Metelli, and Marcello Restelli. "Information-theoretic regret bounds for bandits with fixed expert advice." In 2023 IEEE Information Theory Workshop (ITW), pp. 30-35. IEEE, 2023.

- [Technical Novelty] The obtained results are, in my opinion, quite expected and this is not necessarily an issue. For the $\log T$ regret bound, as the authors acknowledge, this can be obtained by running standard UCB on $Ext(\mathcal{P})$. For the constant regret since the optimal distribution is played for an expected number of times that is linear in $T$, and the optimal distribution plays all the arms with a non-zero probability, we are like in an "expert setting" since playing the optimal distribution provides information for all the arms. Is this consideration correct? Anyway, I encourage the authors to revise the manuscript to highlight in the main paper the challenges of their theoretical analysis and the elements of technical novelty over the previous works, especially [2], with which I noted several elements of similarity in the analysis.

- [Computing $Ext(\mathcal{P})$] Is computing $Ext(\mathcal{P})$ computationally efficient given a description of $\mathcal{P}$ based on constraints? Can the authors elaborate?

- [Tightness] As far as I understand, the lower bound of Theorem 2 provides a lower bound that matches the $\log T$ dependence of the upper bound of Theorem 1. Is there an analogous showing the tightness of the scenario in which the proposed algorithm suffers constant regret? More in general, apart from the dependence on $T$ all lower and upper bounds do not provide the explicit dependence on (i) suboptimality gaps; (iii) number of arms/number of points in $Ext(\mathcal{P})$; (ii) constants. This does not allow to fully judge their tightness.

- [Presentation] Section 3 seems to include several heterogeneous contributions. It starts with the lower bound but then the role and significance of Corollary 1 and Proposition  1 are not fully clear. Can the authors clarify?

- [Non-polytope $\mathcal{P}$] The case in which $\mathcal{P}$ is a general convex set (not a polytope) is just addressed by considering a specific example of  $\mathcal{P}$. It is not clear how much this example is general or how these considerations/methods can be generalized beyond the example.

**Minor Issues**

- The paper lacks a Conclusion section
- Assumption 1: I think the second $E(\zeta)$ should be replaced with $E(\zeta')$

---

### Review · Reviewer_SArx · 2024-05-13

**Summary Of Contributions:**

This paper studies diversity-preserving multi-armed bandits. The authors design a UCB algorithm which enjoys bounded regret when diversity is desirable. The paper also present the regret lower bounds for the mean-unbounded cases.

**Audience:**

Yes

**Claims And Evidence:**

Yes

**Requested Changes:**

1. The writing can be improved, see weaknesses 3.
2. In section 3, the authors may present an example application which is mean-unbounded.
3. The paper may provide some empirical results of the proposed methods. I would like to see the empirical comparisons among  the proposed method, Celis et al (2019) and Liu et al. (2022).

**Strengths And Weaknesses:**

Strengths:

Fairness is an important and hot topic in machine learning area. The studied problem is interesting. The proposed algorithm achieves bounded regret when diversity is desirable.

Weaknesses:

1. The regret lower bound only works for the mean-unbounded cases, which is less common than mean-bouned cases in real applications.
2. The paper does not provide any empirical results for the proposed methods.
3. The main paper is not self-contained. I think the closed-form regret bounds should be presented in the main paper. Also, I think the authors may provide some discussion on the closed-form bound. For example, does the regret bound in Thm 1 match that of vanilla UCB when the polytope is $[0,1]^K$? (In this case, the proposed algorithm seems exactly the same as the vanilla UCB)

---

> ### Author Response · Authors · 2024-05-17
> **Answer: First Part**
>
> We thank the reviewer for raising these points.
>
> **Weakness #1 / Requested Change #2: Mean-unbounded cases**
>
> Indeed, the lower bound of Theorem 2 only holds for mean-unbounded models (while the upper bounds of Theorem 1 hold both for mean-bounded and mean-unbounded models). However, Proposition 1 exactly suggests that building a general lower-bound theory in the mean-bounded case is difficult, or maybe even impossible. We therefore do not see this difficulty or impossibility as a weakness of our approach, but as an intrinsic (and to us, unexpected) limitation of the setting introduced by Celis et al. (2019).
>
> For the motivating problem of ad diversity suggested by Celis et al. (2019), rewards are indeed bounded. We will follow the suggestion of the reviewer and also add an example with mean-unbounded models (that are sub-Gaussian). Admittedly, they are less common in real-world applications. One immediate idea is the model with distributions of the form: some mean $\mu \geq 0$ plus some (possibly wide-spread) sub-Gaussian centered distribution $\nu$. This would model, e.g., in finance applications, returns of unknown level but with known shape. The diversity-preserving constraint could come from the necessity of putting all investments at a given round in a single asset while considering safe allocation policies that put, in expectation, a fraction of the capital in all assets.
>
>
> **Weakness #2 / Requested Change #3: Simulations**
>
> Our contribution was mostly thought to be theoretical, but we are happy to illustrate the theory, e.g., finite regret achieved in the case when the optimal distribution lies in the relative interior of the simplex. We believe that a more extensive empirical comparison would be interesting but would be out of the scope we targeted for this contribution.
>
> We provide some  empirical results in the supplementary PDF material (together with the corrections for the typos in the proof of Theorem 1) --- namely, a figure including simulations we conducted on synthetic data, evaluating our strategy [diversity-preserving UCB, abbreviated DivP-UCB below] and Algorithm 2 [Constrained-L1-OFUL] of Celis et al. (2019)
>
> We compare these algorithms on a family of problems with a fixed polytope action set, but with mean reward vectors that induce different optimal arms -- see details below. By our analysis, finite regret is achievable whenever the optimal arm lies inside the relative interior of the simplex. The simulations confirm these theoretical findings, showing that DivP-UCB indeed reaches finite regret whenever it is possible, and that it is always better than Constrained-L1-OFUL in our experiments. Note also that Constrained-L1-OFUL is computationally more costly than DivP-UCB, since it needs to solve 2d linear programs at every time step, instead of a single one for DivP-UCB.
>
> We also wish to add two clarifying remarks regarding the literature:
> - As explained on page 6 of our submission, the algorithm from Liu et al. (2022) essentially reduces to diversity-preserving UCB in our setting (up to constants in the upper confidence bounds). The authors do not provide numerical experiments.
> - Celis et al. (2019) carry a different set of simulations: they compare the cumulative reward of algorithms *as the action set varies* (i.e., for different thresholds in diversity constraints). This is more of an evaluation of the impact of incorporating diversity constraints, whereas we focus on the performance of the bandit algorithms involved.
>
> Detailed experimental setting:
> We take $K=3$ and $T=3000$. The diversity-preserving polytope we consider is the triangle with vertices
> $\underline{p}_1 = (0, 0.2, 0.8) $,
> $\underline{p}_2 = (0.6, 0.2, 0.2) $,
> $\underline{p}_3 = (0, 0.8, 0.2) $,
>
> an we examine mean-value payoffs of the form:
> $\mu_\alpha  = (1/2 - \alpha, 1/3, 1/2 + \alpha)$, for $\alpha \in \{-0.1, 0.1\}$.
> One can show that depending on the value of $\alpha$, the optimal action may be either $\underline{p}_2$ or $\underline{p}_3$.
> Experiments were run in Python, using numpy and cvxpy for solving linear programs.

---

> ### Author Response · Authors · 2024-05-17
> **Answert Second Part**
>
> **Weakness #3 / Requested Change #1: On the writing**
>
> Yes, we agree that the closed-form expression for the regret upper bounds in Theorem 1 are to be found in the appendix. We had done so for the sake of readability but would be ready to incorporate these bounds in the main body if all 3 reviewers find it useful.
>
> However, to us, apart from this (relatively minor) presentation issue, the main paper is self-contained.
>
> Now, regarding the closed-form regret upper bound scaling with ln T, i.e., the bound (12): Yes, the algorithm reduces to vanilla UCB in the case of the polytope of all probability distributions, but the bound (12) following from the general analysis is larger than the vanilla-UCB bound. This is because the $n_{\underline{p}}$ in (10) are larger than the $\ln T/\Delta_a$ that are obtained in the classic analysis. The deep reason behind this is that we targeted generality.
>
> However, let us underline that the $\ln T$ bound of Theorem 1 is rather a 'sanity check' and that the most interesting result in Theorem 1 (in our eyes and in the ones of the other reviewers) is the constant-regret bound.

---

### Decision · Action_Editor_5KVk · 2024-06-18

**Recommendation:** Accept with minor revision

**Comment:**

The paper revisits the framework for diversity-preserving $K$-armed bandits. It presents a UCB algorithm tailored for this setting, offering theoretical guarantees on its performance. The authors provide regret upper and lower bounds and discuss the application of their approach beyond simple polytopes.

The paper presents valuable theoretical contributions to the field of diversity-preserving $K$-armed bandits. While there are some weaknesses, particularly in "empirical validation and completeness of the main paper", the authors' responses indicate a willingness to address these issues. With the suggested revisions, the paper has the potential to make a significant contribution to the field.

**Audience:**

Yes

**Claims And Evidence:**

Yes